# Neural-Guided Symbolic Regression with Asymptotic Constraints

## Abstract

Symbolic regression is a type of discrete optimization problem that involves searching expressions that fit given data points. In many cases, other mathematical constraints about the unknown expression not only provide more information beyond just values at some inputs, but also effectively constrain the search space. We identify the asymptotic constraints of leading polynomial powers as the function approaches $0$ and $\infty$ as useful constraints and create a system to use them for symbolic regression. The first part of the system is a conditional production rule generating neural network which preferentially generates production rules to construct expressions with the desired leading powers, producing novel expressions outside the training domain. The second part, which we call Neural-Guided Monte Carlo Tree Search, uses the network during a search to find an expression that conforms to a set of data points and desired leading powers. Lastly, we provide an extensive experimental validation on thousands of target expressions showing the efficacy of our system compared to exiting methods for finding unknown functions outside of the training set.

## 1 Introduction

The long standing problem of symbolic regression tries to search expressions in large space that fit given data points (Koza & Koza, 1992; Schmidt & Lipson, 2009). These mathematical expressions are much more like discovered mathematical laws that have been an essential part of the natural sciences for centuries. Since the size of the search space increases exponentially with the length of expressions, current search methods can only scale to find expressions of limited length. Moreover, current symbolic regression techniques fail to exploit a key value of mathematical expressions that has traditionally been well used by natural scientists. Symbolically derivable properties such as bounds, limits, and derivatives can provide significant guidance to finding an appropriate expression.

In this work, we consider one such property corresponding to the behavior of the unknown function as it approaches $0$ and $\infty$. Many expressions have a defined leading polynomial power in these limits. For examples when $x \to \infty$, $2x^2 + 5x$ has a leading power of $2$ (because the expression behaves like $x^2$) and $1/x^2 + 1/x$ has a leading power of $-1$. We call these properties "asymptotic constraints" because this kind of property is known *a priori* for some physical systems before the detailed law is derived. For example, most materials have a heat capacity proportional to $T^3$ at low temperatures $T$ and the gravitational field of planets (at distance $r$) should behave as $1/r$ as $r \to \infty$.

Asymptotic constraints not only provide more information about the expression, leading to better extrapolation, but also constrain the search in the desired semantic subspace, making the search more tractable in much larger space. These constraints can *not* be simply incorporated using syntactic restrictions over the grammar of expressions. We present a system to effectively use asymptotic constraints for symbolic regression, which has two main parts. The first is a conditional production rule generating neural network (NN) of the desired polynomial leading powers that generates production rules to construct novel expressions (both syntactically and semantically) and, more surprisingly, generalize to leading powers not in the training set. The second part is a Neural-Guided Monte Carlo Tree Search (NG-MCTS) that uses this NN to probabilistically guide the search at every step to find expressions that fit a set of data points.

Finally, we provide an extensive empirical evaluation of the system compared to several strong baseline techniques. We examine both the NG-MCTS and conditional production rule generating NN

alone. In sharp contrast to almost all previous symbolic regression work, we evaluate our technique on thousands of target expressions and show that NG-MCTS can successfully find the target expressions in a much larger fraction of cases (71%) than other methods (23%) with search space sizes of more than $10^{50}$ expressions.

In summary, this paper makes the following key contributions: 1) We identify asymptotic constraints as important additional information for symbolic regression tasks. 2) We develop a conditional production rule generating NN to learn a distribution over (syntactically-valid) production rules conditioned on the asymptotic constraints. 3) We develop the NG-MCTS algorithm that uses the conditional production rule generating NN to efficiently guide the MCTS in large space. 4) We extensively evaluate our production rule generating NN to demonstrate generalization for leading powers, and show that the NG-MCTS algorithm significantly outperforms previous techniques on thousands of tasks.

## 2 PROBLEM DEFINITION

In order to demonstrate how prior knowledge can be incorporated into symbolic regression, we construct a symbolic space using a context-free grammar $G$:

$$
\begin{aligned}
O &\rightarrow S \\
S &\rightarrow S\text{`+'}T \mid S\text{`−'}T \mid S\text{`*'}T \mid S\text{`/'}T \mid T \\
T &\rightarrow \text{`('}S\text{`)'} \mid \text{`}x\text{'} \mid \text{`}1\text{'}
\end{aligned}
\tag{1}
$$

This expression space covers a rich family of rational expressions, and the size of the space can be further parameterized by a bound on the maximum sequence length. For an expression $f(x)$, the leading power at $x_0$ is defined as $P_{x \to x_0}[f] = p$ s.t. $\lim_{x \to x_0} f(x)/x^p = $ non-zero constant. In this paper, we consider the leading powers at $x_0 \in \{0, \infty\}$ as additional specification.

Let $\mathcal{S}(G, k)$ denote the space of all expressions in the Grammar $G$ with a maximum sequence length $k$. Conventional symbolic regression searches for a desired expression $f(x)$ in the space of expressions $\mathcal{S}(G, k)$ that conforms to a set of data points $\{(x, f(x)) \mid x \in \mathcal{D}_{\text{train}}\}$, i.e. find a $g(x) \in \mathcal{S}(G, k) : \phi(g(x), \mathcal{D}_{\text{train}})$, where $\phi$ denotes the acceptance criterion, usually root mean square error (RMSE). With the additional specification of leading powers $c^{(0)}$ and $c^{(\infty)}$ at 0 and $\infty$, the problem becomes: find a $g(x) \in \mathcal{S}(G, k) : \phi(g(x), \mathcal{D}_{\text{train}}) \wedge (P_{x \to 0}[g] = c^{(0)}) \wedge (P_{x \to \infty}[g] = c^{(\infty)})$.

## 3 CONDITIONAL PRODUCTION RULE GENERATING NEURAL NETWORK

It is difficult to directly incorporate the asymptotic constraints as syntactic restrictions over the grammar. We, therefore, develop a neural architecture that learns to generate production rule in the grammar conditioned on the given asymptotic constraints.

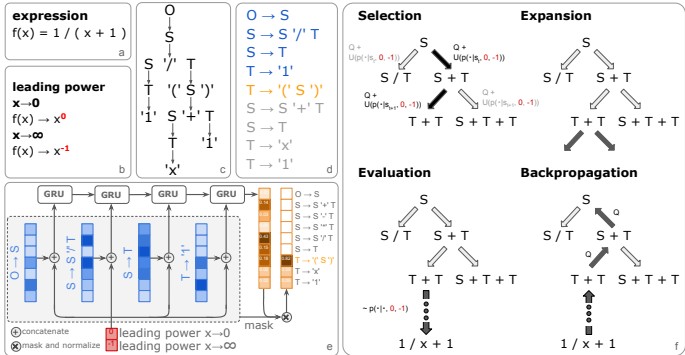

Figure 1: **Overview.** (a) Exemplary expression. (b) Leading powers of $1/(x + 1)$ at 0 and $\infty$. (c) Parse tree of $1/(x + 1)$. (d) Production rule sequence, the preorder traversal of production rules in the parse tree. (e) Architecture of the model to predict the next production rule from the partial sequence conditioned on desired leading powers. (f) Using (e) to guide MCTS.

Figure 1(a) and (c) show an example of how an expression is parsed as a parse tree by the grammar defined in Eq. (1). The parse tree in Figure 1(c) can be serialized into a production rule sequence $r_1, \ldots, r_L$ by a preorder traversal (Figure 1(d)), where $L$ denotes the length of the production rule sequence. Figure 1(b) shows the leading powers of the exemplary expression in Figure 1(a). The conditional distribution of an expression is parameterized as a sequential model

$$p_\theta(f|c^{(0)}, c^{(\infty)}) = \prod_{t=1}^{L-1} p_\theta(r_{t+1}|r_1, \ldots, r_t, c^{(0)}, c^{(\infty)}). \tag{2}$$

We build a NN (as shown in Figure 1(e)) to predict the next production rule $r_{t+1}$ from a partial sequence $r_1, \ldots, r_t$ and conditions $c^{(0)}, c^{(\infty)}$. During training, each expression in the training set is first parsed as a production rule sequence. Then a partial sequence of length $t \in \{1, \ldots, L-1\}$ is sampled randomly as the input and the $(t+1)$-th production rule is selected as the output (see blue and orange text in Figure 1(d)). Each production rule of the partial sequence is represented as an embedding vector of size 10. The conditions are concatenated with each embedding vector. This sequence of embedding vectors are fed into a bidirectional Gated Recurrent Units (GRU) (Cho et al., 2014) with 1000 units. A softmax layer is applied to the final output of GRU to obtain the raw probability distribution over the next production rules in Eq. (1).

Note that not all the production rules are grammatically valid as the next production rule for a given partial sequence. The partial sequence is equivalent to a partial parse tree. The next production rule expands the leftmost non-terminal symbol in the partial parse tree. For the partial sequence colored in blue in Figure 1(d), the next production rule expands non-terminal symbol $T$, which constrains the next production rule to only those with left-hand-side symbol $T$. We use a stack to keep track of non-terminal symbols in a partial sequence as described in GVAE (Kusner et al., 2017). A mask of valid production rules is computed from the input partial sequence. This mask is applied to the raw probability distribution and the result is normalized to 1 as the output probability distribution. The training loss is calculated as the cross entropy between the output probability distribution and the next target production rule. It is trained from partial sequences sampled from expressions in the training set using validation loss for early stopping.[1]

## 4 NEURAL-GUIDED MONTE CARLO TREE SEARCH

We now briefly describe the NG-MCTS algorithm that uses the conditional production rule generating NN to guide the symbolic regression search. The discrepancy between the best found expression $g(x)$ and the desired $f(x)$ is evaluated on data points and leading powers. The error on data points is measured by RMSE $\Delta g_{\{\cdot\}} = \sqrt{\sum_{x \in \mathcal{D}_{\{\cdot\}}} (f(x) - g(x))^2 / |\mathcal{D}_{\{\cdot\}}|}$ on training points $\mathcal{D}_{\text{train}}$ : $\{1.2, 1.6, 2.0, 2.4, 2.8\}$, points in interpolation region $\mathcal{D}_{\text{interpolation}}$ : $\{1.4, 1.8, 2.2, 2.6\}$ and points in extrapolation region $\mathcal{D}_{\text{extrapolation}}$ : $\{5, 6, 7, 8, 9\}$. The error on leading powers is measured by sum of absolute errors at 0 and $\infty$, $\Delta P[g] = |P_{x \to 0}[f] - P_{x \to 0}[g]| + |P_{x \to \infty}[f] - P_{x \to \infty}[g]|$. The default choice of objective function for symbolic regression algorithms is $\Delta g_{\text{train}}$ alone. With additional leading powers constraint, the objective function can be defined as $\Delta g_{\text{train}} + \Delta P[g]$, which minimizes both the RMSE on the training points and the absolute difference of the leading powers.

Most symbolic regression algorithms are based on EA (Schmidt & Lipson, 2009), where it is nontrivial to incorporate our conditional production rule generating NN to guide the generation strategy in a step-by-step manner, as the mutation and cross-over operators perform transformations on fully completed expressions. However, it is possible to incorporate a probability distribution over expressions in many heuristic search algorithms such as Monte Carlo Tree Search (MCTS). MCTS is a heuristic search algorithm that has been shown to perform exceedingly well in problems with large combinatorial space, such as mastering the game of Go (Silver et al., 2016) and planning chemical syntheses (Segler et al., 2018). In MCTS for symbolic regression, a partial parse tree sequence $r_1, \ldots, r_t$ can be defined as a state $s_t$ and the next production rule is a set of actions $\{a\}$. In the selection step, we use a variant of the PUCT algorithm (Silver et al., 2016; Rosin, 2011) for exploration. For MCTS, the prior probability distribution $p(a_i|s_t)$ is uniform among all valid actions.

We develop NG-MCTS by incorporating the conditional production rule generating NN into MCTS for symbolic regression. Figure 1(f) presents a visual overview of NG-MCTS. In particular, the prior

---

[1]Model implemented in TensorFlow (Abadi et al., 2016) and available in submitted materials.

Table 1: **Results of symbolic regression methods.** Search expressions in holdout sets $M[f] \leq 4$, $M[f] = 5$ and $M[f] = 6$ with data points on $\mathcal{D}_{\text{train}}$ and / or leading powers $P_{x\to 0}[f]$ and $P_{x\to\infty}[f]$. The options are marked by on ($\sqrt{}$), off ($\times$) and not available (–). If the RMSEs of the best found expression $g(x)$ in interpolation and extrapolation are both smaller than $10^{-9}$ and $\Delta P[g] = 0$, it is *solved*. If $g(x)$ is non-terminal or $\infty$, it is *invalid*. *Hard* includes expressions in the holdout set which are not solved by any of the six methods. The medians of $\Delta g_{\text{train}}$, $\Delta g_{\text{int.}}$, $\Delta g_{\text{ext.}}$ and the median absolute errors of leading powers $\Delta P[g]$ for hard expressions are reported.

| $M[f]$ | Method | Neural Guided | Objective Function | | Solved | Invalid | Hard | | | | |
| --- | --- | --- | --- | --- | --- | --- | --- | --- | --- | --- | --- |
| | | | $\mathcal{D}_{\text{train}}$ | $P_{x\to 0,\infty}[f]$ | Percent | Percent | Percent | $\Delta g_{\text{train}}$ | $\Delta g_{\text{int.}}$ | $\Delta g_{\text{ext.}}$ | $\Delta P[g]$ |
| $\leq 4$ | MCTS | $\times$ | $\sqrt{}$ | $\times$ | 0.54% | 2.93% | | 0.728 | 0.598 | 0.723 | 3 |
| | MCTS (PW-only) | $\times$ | $\times$ | $\sqrt{}$ | 0.24% | **0.00%** | | – | 2.069 | 2.823 | 1 |
| | MCTS + PW | $\times$ | $\sqrt{}$ | $\sqrt{}$ | 0.20% | 0.39% | 23.66% | 0.967 | 0.836 | 0.541 | 2 |
| | NG-MCTS | $\sqrt{}$ | $\sqrt{}$ | $\sqrt{}$ | **71.22%** | **0.00%** | | 0.225 | 0.194 | **0.084** | **0** |
| | EA | – | $\sqrt{}$ | $\times$ | 12.83% | 3.32% | | **0.186** | **0.162** | 0.358 | 3 |
| | EA + PW | – | $\sqrt{}$ | $\sqrt{}$ | 23.37% | 0.44% | | 0.376 | 0.322 | 0.152 | **0** |
| | GVAE‡ | – | $\sqrt{}$ | $\times$ | 10.00% | 0.00% | 90.00% | 0.217 | 0.159 | 0.599 | 2 |
| | GVAE + PW‡ | – | $\sqrt{}$ | $\sqrt{}$ | 10.00% | 0.00% | 90.00% | 0.386 | 0.324 | 0.056 | 0 |
| $= 5$ | MCTS | $\times$ | $\sqrt{}$ | $\times$ | 0.00% | 3.90% | | 0.857 | 0.738 | 0.950 | 5 |
| | MCTS (PW-only) | $\times$ | $\times$ | $\sqrt{}$ | 0.00% | **0.00%** | | – | 1.890 | 1.027 | 3 |
| | MCTS + PW | $\times$ | $\sqrt{}$ | $\sqrt{}$ | 0.10% | 3.50% | 58.00% | 1.105 | 0.914 | 0.600 | 4 |
| | NG-MCTS | $\sqrt{}$ | $\sqrt{}$ | $\sqrt{}$ | **32.10%** | **0.00%** | | 0.247 | 0.229 | **0.020** | **0** |
| | EA | – | $\sqrt{}$ | $\times$ | 2.90% | 4.20% | | **0.227** | **0.204** | 0.155 | 4 |
| | EA + PW | – | $\sqrt{}$ | $\sqrt{}$ | 9.20% | 2.30% | | 0.366 | 0.365 | 0.109 | **0** |
| | GVAE‡ | – | $\sqrt{}$ | $\times$ | 0.00% | 0.00% | 100.00% | 0.233 | 0.259 | 0.164 | 4 |
| | GVAE + PW‡ | – | $\sqrt{}$ | $\sqrt{}$ | 3.33% | 0.00% | 96.67% | 0.649 | 0.565 | 0.383 | 2 |
| $= 6$ | MCTS | $\times$ | $\sqrt{}$ | $\times$ | 0.00% | 6.33% | | 1.027 | 0.819 | 0.852 | 6 |
| | MCTS (PW-only) | $\times$ | $\times$ | $\sqrt{}$ | 0.00% | **0.00%** | | – | 2.223 | 7.145 | 4 |
| | MCTS + PW | $\times$ | $\sqrt{}$ | $\sqrt{}$ | 0.08% | 7.33% | 71.25% | 1.228 | 1.051 | 0.891 | 4 |
| | NG-MCTS | $\sqrt{}$ | $\sqrt{}$ | $\sqrt{}$ | **17.33%** | 0.17% | | 0.236 | 0.209 | **0.008** | **0** |
| | EA | – | $\sqrt{}$ | $\times$ | 1.25% | 5.08% | | **0.219** | **0.191** | 0.084 | 5 |
| | EA + PW | – | $\sqrt{}$ | $\sqrt{}$ | 4.92% | 6.92% | | 0.329 | 0.285 | 0.047 | **0** |
| | GVAE‡ | – | $\sqrt{}$ | $\times$ | 0.00% | 0.00% | 100.00% | 0.260 | 0.206 | 0.037 | 5 |
| | GVAE + PW‡ | – | $\sqrt{}$ | $\sqrt{}$ | 0.00% | 0.00% | 100.00% | 0.595 | 0.436 | 0.087 | 3 |

‡ Evaluated on the subset of holdout sets. *Hard* are unsolved expressions in the subset (Appendix H).

probability distribution $p(a_i|s_t, c^{(0)}, c^{(\infty)})$ is computed by our conditional production rule generating NN on the partial sequence and the desired conditions. We run MCTS for 500 simulations for each desired expression $f(x)$. The exploration strength is set to 50 and the production rule sequence length limit is set to 100. The total number of expressions in this combinatorial space is $3 \times 10^{93}$. We run evolutionary algorithm (EA) (Appendix G) and grammar variational autoencoder (GVAE) (Kusner et al., 2017) (Appendix H) with comparable computational setup for comparison.

## 5 EVALUATION

**Dataset and Search Space:** We denote the leading power constraint as a pair of integers $(P_{x\to 0}[f], P_{x\to\infty}[f])$ and define the complexity of a condition by $M[f] = |P_{x\to 0}[f]| + |P_{x\to\infty}[f]|$. Obviously, expressions with $M[f] = 4$ are more complicated to construct than those with $M[f] = 0$. We create a dataset balanced on each condition, as described in Appendix A. The conditional production rule generating NN is trained on 28837 expressions and validated on 4095 expressions, both with $M[f] \leq 4$. The training expressions are sampled *sparsely*, which are only $10^{-23}$ of the expressions within 31 production rules. Symbolic regression tasks are evaluated on 4250 expressions in holdout sets with $M[f] \leq 4, = 5, = 6$. The challenges are: 1) conditions $M[f] = 5, 6$ do not exist in training set; 2) the search spaces of $M[f] = 5, 6$ are $10^7$ and $10^{11}$ times larger than the training set.

### 5.1 EVALUATION OF SYMBOLIC REGRESSION TASKS

We now present the evaluation of our NG-MCTS method, where each step in the search is guided by the conditional production rule generating NN. Recent developments of symbolic regression methods (Kusner et al., 2017; Sahoo et al., 2018) compare methods on only a few expressions, which may cause the performance to depend on random seed for initialization and delicate tuning. To mitigate this, we apply different symbolic regression methods to search for expressions in holdout sets with thousands of expressions and compare their results in Table 1. Additional comparison on a subset of holdout sets is reported in Appendix H.

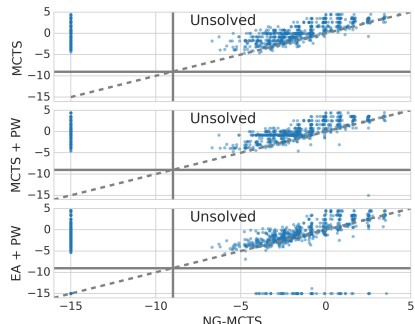
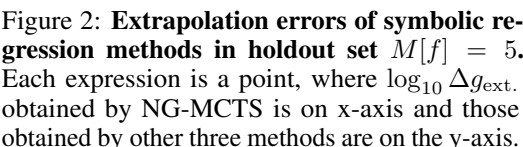
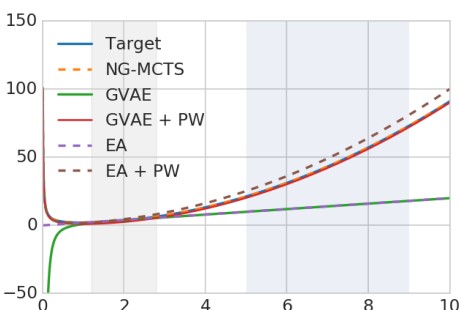

Figure 2: **Extrapolation errors of symbolic regression methods in holdout set** $M[f] = 5$. Each expression is a point, where $\log_{10} \Delta g_{\text{ext.}}$ obtained by NG-MCTS is on x-axis and those obtained by other three methods are on the y-axis.

Figure 3: **Plot of force field expressions found by each method.** Grey area is the region to compute interpolation error $\Delta g_{\text{int.}}$ and light blue area is the region to compute extrapolation error $\Delta g_{\text{ext.}}$.

We first discuss the results for holdout set $M[f] \leq 4$. Conventional symbolic regression only fits on the data points $\mathcal{D}_{\text{train}}$. EA solves $12.83\%$ expressions, while MCTS only solves $0.54\%$ expressions. This suggests that compared to EA, MCTS is not efficient in searching a large space with limited number of simulations. The median errors $\Delta P[g]$ are both 3 for *hard* expressions (expressions unsolved by all methods), which are large as maximum $M[f]$ for this set is 4.

In order to examine the effect of leading powers, we use leading powers alone in MCTS (PW-ONLY). The median of $\Delta P[g]$ for hard expressions is reduced to 1 but the medians of $\Delta g_{\text{int.}}$ and $\Delta g_{\text{ext.}}$ are significantly higher. We then add leading powers to the objective function together with data points. MCTS + PW does not have a notable difference to MCTS. However, EA + PW improves solved expressions to $23.37\%$ and $\Delta P[g]$ of hard expressions is 0. This indicates adding leading power constraints in the objective function is helpful for symbolic regression. Most importantly, we observe step-wise guidance of NN conditioned on leading powers can lead to even more significant improvements compared to adding them in the objective function. NG-MCTS solves $71.22\%$ expressions in the holdout set, three times over the best EA + PW. Note that both MCTS + PW and EA + PW have access to the same input information. Although EA has the lowest medians of $\Delta g_{\text{train}}$ and $\Delta g_{\text{int.}}$, NG-MCTS is only slightly worse. On the other hand, NG-MCTS outperforms on $\Delta g_{\text{ext.}}$ and $\Delta P[g]$, which indicates that step-wise guidance of leading powers helps to generalize better in extrapolation than all other methods.

We also apply the aforementioned methods to search expressions in holdout sets $M[f] = 5$ and $M[f] = 6$. The percentage of solved expressions decreases as $M[f]$ increases as larger $M[f]$ requires learning expressions with more complex syntactic structure. The median of $\Delta P[g]$ also increases with larger $M[f]$ for the other methods, but the value for NG-MCTS is *always* zero. This demonstrates that our NN model is able to successfully guide the NG-MCTS even for leading powers not appearing in the training set. Due to the restriction on the computational time of Bayesian optimization for GVAE, we evaluate GVAE + PW on a subset of holdout sets (Appendix H). GVAE+ PW fails in holdout sets $M[f] = 5, 6$. Overall, NG-MCTS still significantly outperforms other methods in solved percentage and extrapolation. Figure 2 compares $\Delta g_{\text{ext.}}$ for each expression among different methods in holdout set $M[f] = 5$. The upper right cluster in each plot represents expressions unsolved by both methods. Most of the plotted points are above the 1:1 line (dashed), which shows that NG-MCTS outperforms the others for most unsolved expressions in extrapolation. Examples of expressions solved by NG-MCTS but unsolved by EA + PW and vice versa are presented in Appendix J. We also perform similar experiments with Gaussian noise on $\mathcal{D}_{\text{train}}$ and NG-MCTS still outperforms all other methods ( Appendix K).

## 5.2 CASE STUDY: FORCE FIELD POTENTIAL

Molecular dynamics simulations (Alder & Wainwright, 1959) study the dynamic evolution of physical systems, with extensive applications in physics, quantum chemistry, biology and material science. The interaction of atoms or coarse-grained particles (Kmiecik et al., 2016) is described by potential energy function called force field, which is derived from experiments or computations of quantum mechanics algorithms. Typically, researchers know the interactions in short and long ranges, which are examples

Table 2: Results of force field expressions found by each method.

| Method | Expression Found | $\Delta g_{\text{train}}$ | $\Delta g_{\text{int.}}$ | $\Delta g_{\text{ext.}}$ | $\Delta P[g]$ |
|---|---|---|---|---|---|
| NG-MCTS | $1 - x + (1/x) + x \times x$ | **0.00** | **0.00** | **0.00** | **0** |
| GVAE | $(x) - (1/x)/(x \times x/x) + x$ | 0.47 | 0.29 | 34.9 | 2 |
| GVAE + PW | $((1/x) - x + x) - ((1 - x) \times x)$ | 1.0 | 1.0 | 1.0 | 0 |
| EA | $(x + x)$ | 0.52 | 0.46 | 34.8 | 3 |
| EA + PW | $((1/x) + (x \times x))$ | 1.15 | 1.10 | 6.16 | 0 |

of asymptotic constraints. We propose a force field potential $U(x) = 1/x + x + (x - 1)^2$ with Coulomb interaction, uniform electric field and harmonic interaction. Assuming the true potential is unknown, the goal is to discover this expression. As a physical potential, besides values at $\mathcal{D}_{\text{train}} : \{1.2, 1.6, 2.0, 2.4, 2.8\}$, researchers also know the short ($x \to 0$) and long range ($x \to \infty$) behaviors as leading powers. Table 2 shows the expressions found by NG-MCTS, GVAE, GVAE + PW, EA and EA + PW, which are plotted in Figure 3. NG-MCTS can find the desired expression. The second best method is GVAE + PW, differing by a constant of 1 from the true expression.

## 5.3 Evaluation of Conditional Production Rule Generating NN

NG-MCTS significantly outperforms other methods on searching expressions in large space. To better understand the effective guidance from NN, we demonstrate its ability to generate syntactically and semantically novel expressions given desired conditions. In order to examine the NN alone, we directly sample from the model by Eq. (2) instead of using MCTS. The model predicts the probability distribution over the next production rules from the starting rule $r_1 : O \to S$ and desired condition $(c^{(0)}, c^{(\infty)})$. The next production rule is sampled from distribution $p_\theta(r_2|r_1, c^{(0)}, c^{(\infty)})$ and then appended to $r_1$. Then $r_3$ is sampled from $p_\theta(r_3|r_1, r_2, c^{(0)}, c^{(\infty)})$ and appended to $[r_1, r_2]$. This procedure is repeated until $[r_1, \ldots, r_L]$ form a parse tree where all the leaf nodes are terminal, or the length of generated sequence reaches the prespecified limit, which is set to 100 for our experiments.

**Baseline Models**  We compare NN with a number of baseline models that provide a probability distribution over the next production rules. All these distributions are masked by the valid production rules computed from the partial sequence before sampling. For each desired condition within $|P_{x \to 0}[f]| \leq 9$ and $|P_{x \to \infty}[f]| \leq 9$, $k = 100$ expressions are generated.

We consider the following baseline models: i) *Neural Network No Condition (NNNC)*: same setup as NN except no conditioning on leading powers, ii) *Random*: uniform distribution over valid next production rules, iii) *Full History (FH)*: using full partial sequence and conditions to sample next production rule from its empirical distribution, iv) *Full History No Condition (FHNC)*, v) *Limited History (LH) (l)*: sampling next production rule from its empirical distribution given only the last $l$ rules in the partial sequence, and vi) *Limited History No Condition (LHNC) (l)*. Note that the aforementioned empirical distributions are derived from the training set $\{f\}$. For limited history models, if $l$ exceeds the length of the partial sequence, we instead take the full partial sequence. More details about the baseline models can be found in Appendix B.

**Metrics**  We propose four metrics to evaluate the performance. For each condition $(c^{(0)}, c^{(\infty)})$, $k$ expressions $\{g_i\}$ are generated from model $p_\theta(f|c^{(0)}, c^{(\infty)})$. i) *Success Rate*: proportion of generated expressions with leading powers $(P_{x \to 0}[g_i], P_{x \to \infty}[g_i])$ that match the desired condition, ii) *Mean L1-distance*: the mean L1-distance between $(P_{x \to 0}[g_i], P_{x \to \infty}[g_i])$ and $(c^{(0)}, c^{(\infty)})$, iii) *Syntactic Novelty Rate*: proportion of generated expressions that satisfy the condition and are syntactically novel (no syntactic duplicate of generated expression in the training set), and iv) *Semantic Novelty Rate*: proportion of expressions satisfying the condition and are semantically novel (the expression obtained after normalizing the generated expression does not exist in the training set). For example, expressions $x + 1$ and $(1) + x$ are syntactically different, but semantically duplicate. We use simplify function in SymPy (Meurer et al., 2017) to normalize expressions. To avoid inflating the rates of syntactic and semantic novelty, we only count the number of unique syntactic and semantic novelties in terms of their expressions and simplified expressions, respectively.

**Quantitative Evaluation**  Table 3 compares the model performance of baseline and NN models measured by different metrics. We define $M[f] \leq 4$ as *in-sample condition region* and $M[f] > 4$ as *out-of-sample condition region*. In both regions, the generalization ability of the model is reflected by the number of syntactic and semantic novelties it generates, not just the number of successes. For

Table 3: Metrics for conditional production rule generating NN and baseline models.

| Model | $M[f] \leq 4$ | | $M[f] > 4$ | | | | $M[f] \leq 4$ | $= 5$ | $= 6$ | $=7$ |
|---|---|---|---|---|---|---|---|---|---|---|
| | Syn (%) | Sem (%) | Total Num Expressions | | | Num Conditions with Suc | Mean L1-Dist | | | |
| | | | Suc | Syn | Sem | | | | | |
| NN | **35.0** | **2.7** | **1465** | **1416** | **1084** | **115** | 0.8 | **1.4** | **2.5** | **4.3** |
| NNNC | 1.8 | 0.2 | 7 | 7 | 7 | 7 | 4.0 | 5.6 | 6.5 | 7.5 |
| Random | 0.7 | 0.0 | 0 | 0 | 0 | 0 | 10.9 | 11.7 | 12.4 | 12.6 |
| FH | 0.0 | 0.0 | 0 | 0 | 0 | 0 | **0.0** | 18.0 | 18.0 | 18.0 |
| FHNC | 0.0 | 0.0 | 0 | 0 | 0 | 0 | 4.2 | 5.7 | 6.6 | 7.5 |
| LH (2) | 5.0 | 1.1 | 0 | 0 | 0 | 0 | 3.1 | 18.0 | 18.0 | 18.0 |
| LH (4) | 10.3 | 2.1 | 0 | 0 | 0 | 0 | 2.5 | 18.0 | 18.0 | 18.0 |
| LH (8) | 14.6 | 2.4 | 0 | 0 | 0 | 0 | 1.8 | 18.0 | 18.0 | 18.0 |
| LH (16) | 1.2 | 0.1 | 0 | 0 | 0 | 0 | 0.1 | 18.0 | 18.0 | 18.0 |
| LHNC (2) | 1.4 | 0.3 | 7 | 7 | 7 | 6 | 4.2 | 5.6 | 6.9 | 7.5 |
| LHNC (4) | 1.2 | 0.3 | 6 | 6 | 6 | 6 | 3.9 | 5.7 | 6.4 | 7.3 |
| LHNC (8) | 1.5 | 0.3 | 8 | 8 | 8 | 6 | 4.2 | 5.9 | 6.7 | 7.6 |
| LHNC (16) | 0.2 | 0.1 | 3 | 3 | 3 | 2 | 4.3 | 5.7 | 6.6 | 7.5 |

example, FH behaves as a look-up table based method (i.e., sampling from the training set) so it has $100\%$ success rate in in-sample condition region. However, it is not able to generate any novel expressions. NN has the best performance on the syntactic and semantic novelty rates in both the in-sample ($35\%$ and $2.7\%$) and out-of-sample ($1416$ and $1084$) condition regions by a significant margin. This indicates the generalization ability of the model to generate unseen expressions matching a desired condition. It is worth pointing out that NNNC performs much worse than NN, which indicates that NN is not only learning the distribution of expressions in the dataset, but instead is also learning a conditional distribution to map leading powers to the corresponding expression distributions.

Furthermore, the L1-distance measures the deviation from the desired condition when not matching exactly. NN has the least mean L1-distance in the out-of-sample condition region. This suggests that for the unmatched expressions, NN prefers expressions with leading powers closer to the desired condition than all other models. NN outperforms the other models not only on the metrics aggregated over all conditions, but also for individual conditions. Figure 4 shows the metrics for NN and LHNC (8) on each condition. NN performs better in the in-sample region (inside the red boundary) and also generalizes to more conditions in the out-of-sample region (outside the red boundary).

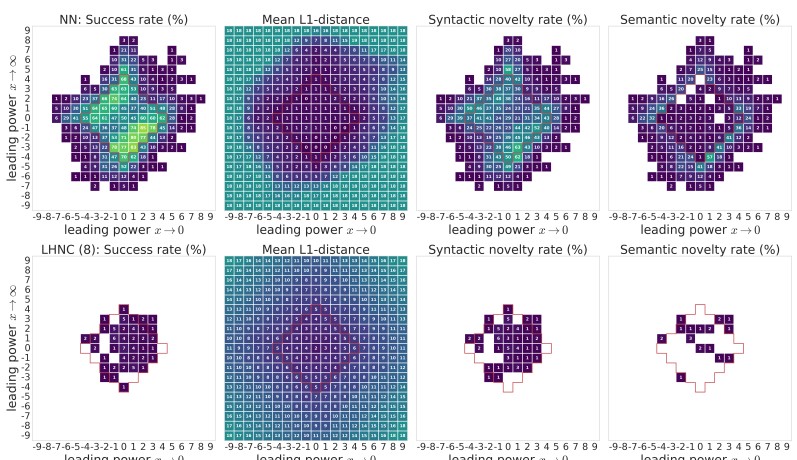

Figure 4: **Visualizing metrics for conditional production rule generating NN and LHNC (8) on each condition within** $|P_{x\to 0}[f]| \leq 9$ **and** $|P_{x\to\infty}[f]| \leq 9$**.** Conditions with $M[f] \leq 4$ are inside the red boundary and points with 0 value are left blank.

**Qualitative Evaluation** To better comprehend the learned NN model and its generative behavior, we also perform a task of expression completion given a structure template of the form $1/\square - \square$ and a variety of desired conditions in Table 4. For each condition, 1000 expressions are generated by NN and the probability of each syntactically unique expression is computed from its occurrence. We first start with $c^{(0)} = 0, c^{(\infty)} = 1$. The completed expression $g(x)$ is required to be a nonzero constant as $x \to 0$ and $g(x) \to x$ as $x \to \infty$. The top three probabilities are close since this task is relatively easy to complete. We then repeat the task on $c^{(0)} = -1, c^{(\infty)} = 1$ and $c^{(0)} = -2, c^{(\infty)} = 2$, which

are still in the in-sample condition region and the model can still complete expressions that match the desired conditions. We also show examples of $c^{(0)} = -3, c^{(\infty)} = 2$, which is in the out-of-sample condition region. Note that to match condition $c^{(\infty)} = -3$, more complicated completion such as $1/(x * (x * x))$ has to be constructed by the model. Even in this challenging case, the model can still generate some expressions that match the desired condition. We also show examples of the syntactic novelties learned by model in Appendix L.

## 6 RELATED WORK

**Symbolic Regression:** Schmidt & Lipson (2009) present a symbolic regression technique to learn natural laws from experimental data. The symbolic space is defined by operators $+$, $-$, $*$, $/$, sin, cos, constants, and variables. An expression is represented as a graph, where intermediate nodes represent operators and leaves represent coefficients and variables. The EA varies the structures to search new expressions using a score that accounts for both accuracy and the complexity of the expression. This approach has been further used to get empirical expressions in electronic engineering (Ceperic et al., 2014), water resources (Klotz et al., 2017), and social science (Truscott & Korns, 2014). GVAE (Kusner et al., 2017) was recently proposed to learn a generative model of structured arithmetic expressions and molecules, where the latent representation captures the underlying structure. This model was further shown to improve a Bayesian optimization based method for symbolic regression.

Table 4: Examples of top-3 probable expression completions for different desired conditions.

| $M[f]$ | $c^{(0)}$ | $c^{(\infty)}$ | Expression | Probability | Match |
|---|---|---|---|---|---|
| 1 | 0 | 1 | $1/\boxed{(1+1)} - \boxed{x}$ | 7.8% | ✓ |
| | | | $1/\boxed{(1+x)} - \boxed{x}$ | 7.7% | ✓ |
| | | | $1/\boxed{x} - \boxed{x}$ | 7.0% | × |
| 2 | -1 | 1 | $1/\boxed{(x+x)} - \boxed{x}$ | 17.3% | ✓ |
| | | | $1/\boxed{(x)} - \boxed{x}$ | 12.0% | ✓ |
| | | | $1/\boxed{x} - \boxed{x}$ | 6.8% | ✓ |
| 4 | -2 | 2 | $1/\boxed{(x*x)} - \boxed{(x*x)}$ | 48.5% | ✓ |
| | | | $1/\boxed{x} - \boxed{(x*x)}$ | 12.5% | × |
| | | | $1/\boxed{(x*x)} - \boxed{x}$ | 7.6% | × |
| 5 | -3 | 2 | $1/\boxed{(x*x)} - \boxed{(x*x)}$ | 29.5% | × |
| | | | $1/\boxed{(x*x)} - \boxed{(x*(x*x))}$ | 19.3% | × |
| | | | $1/\boxed{(x*(x*x))} - \boxed{(x*x)}$ | 12.9% | ✓ |

Similar to these approaches, most other approaches search for expressions from scratch using only data points (Schmidt & Lipson, 2009; Ramachandran et al., 2017; Ouyang et al., 2018) without other symbolic constraints about the desired expression. Abu-Mostafa (1994) suggests incorporating prior knowledge of a similar form to our property constraints, but actually implements those priors by adding additional data points and terms in the loss function.

**Neural Program Synthesis:** Program synthesis is the task of learning programs in a domain-specific language (DSL) that satisfy a given specification (Gulwani et al., 2017). It is closely related to symbolic regression, where the DSL can be considered as a grammar defining the space of expressions. Some recent works use neural networks for learning programs (Devlin et al., 2017; Balog et al., 2016; Parisotto et al., 2017; Vijayakumar et al., 2018). RobustFill (Devlin et al., 2017) trains an encoder-decoder model that learns to decode programs as a sequence of tokens given a set of input-output examples. For more complex DSL grammars such as Karel that consists of nested control-flow, an additional grammar mask is used to ensure syntactic validity of the decoded programs (Bunel et al., 2018). However, these approaches only use examples as specification. Our technique can be useful to guide search for programs where the program space is defined using grammars such as SyGuS with additional semantic constraints other than only examples (Alur et al., 2013; Si et al., 2018).

## 7 CONCLUSION

We present a two-step framework of first learning a neural network of the relation between syntactic structure and leading powers and then using that to guide MCTS for efficient search. This framework is evaluated in the context of symbolic regression and focused on the leading power properties on thousands of desired expressions, a much larger evaluation set than benchmarks considered in existing literature. We plan to further extend the applicability of this framework to cover other symbolically derivable properties of expressions. Similar modeling ideas could be equally useful in general program synthesis settings, where other properties such as the desired time complexity or maximum control flow nesting could be used as constraints.

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

## A   DATASET

To create an initial dataset, we first enumerate all possible parse trees from

$$O \rightarrow S$$
$$S \rightarrow S\text{`+'}T \mid S\text{`-'}T \mid S\text{`*'}T \mid S\text{`/'}T \mid T$$
$$T \rightarrow \text{`('}S\text{`)'} \mid \text{`}x\text{'} \mid \text{`}1\text{'}$$

within ten production rules. Then we repeat the following downsampling and augmentation operations for four times to expand the dataset for longer expressions and diverse conditions.

**Downsampling**   Expressions are grouped by their simplified expressions computed by SymPy (Meurer et al., 2017). We select 20 shortest expressions in terms of string length from each group. These expressions are kept to ensure syntactical diversity and avoid having too long expressions.

**Augmentation**   For each kept expression, five new expressions are created by randomly replacing one of the 1 or $x$ symbols by $(1/x)$, $(x/(1+x))$, $(x/(1-x))$, $(1/(1+x))$, $(1/(1-x))$, $(1-x)$, $(1+x)$, $(x*x)$, $(x*(1+x))$, $(x*(1-x))$. These five newly created expressions are added back to the pool of kept expressions to form an expanded set of expressions.

After repeating the above operations for four times, we apply the downsampling step again in the end. To make the dataset balanced on each condition, we keep 1000 shortest expressions in terms of string length for each condition. In this way, we efficiently create an expanded dataset which is not only balanced on each condition but also contains a large variety of expressions with much longer length than the initial dataset. Compared to enumerating all possible expressions given the maximum length, the created dataset is much sparser and smaller.

For each pair of integer leading powers satisfying $M[f] \leq 4$, 1000 shortest expressions are selected to obtain 41000 expressions in total. They are randomly split into three sets. The first two are training (28837) and validation (4095) sets. For the remaining expressions, 50 expressions with unique simplified expressions are sampled from each condition for $M[f] \leq 4$, to form a holdout set with 2050 expressions. In the same way, we also create a holdout set of 1000 expressions for $M[f] = 5$ and 1200 expressions for $M[f] = 6$. These conditions do not exist in training and validation sets.

Table A.1: Minimum, median and maximum of the lengths of the production rule sequence. Space size is the number of possible expressions within the maximum length of the production rule sequence.

| Name | Length | | | Number of Expressions | Space Size |
|---|---|---|---|---|---|
| | Min | Median | Max | | |
| training $M[f] \leq 4$ | 3 | 19 | 31 | 28837 | $2.2 \times 10^{27}$ |
| holdout $M[f] \leq 4$ | 7 | 19 | 31 | 2050 | $2.2 \times 10^{27}$ |
| holdout $M[f] = 5$ | 15 | 27 | 39 | 1000 | $8.9 \times 10^{34}$ |
| holdout $M[f] = 6$ | 11 | 31 | 43 | 1200 | $5.8 \times 10^{38}$ |

Figure A.1 shows the histogram of lengths of the production rule sequence for expressions in the training set and holdout sets. Table A.1 shows the minimum, median and maximum of the lengths of the production rule sequence. The last column, space size, shows the number of possible expressions within the maximum length of the production rule sequence (including those non-terminal expressions). It is computed recursively as follows. Let $N_{*,i}$ denote the number of possible expressions with length $\leq i$ and whose production rule sequences start with symbol $*$, where $*$ can be $O, S$ and $T$. Then we have

$$N_{S,i} = 4 \sum_{p=0}^{i-1} (N_{S,p} \cdot N_{T,i-1-p}) + N_{T,i-1};$$

$$N_{T,i} = N_{S,i-1} + 2,$$

and the initial condition is $N_{S,0} = N_{T,0} = 1$. We can obtain $N_{S,i}$ using the recursive formula. The quantity of interest $N_{O,i} = N_{S,i-1}$ given that the first production rule is pre-defined as $O \rightarrow S$. Holdout sets $M[f] = 5, 6$ not only contain expressions of higher leading powers but also of longer length, which is challenging for generalization both semantically and syntactically.

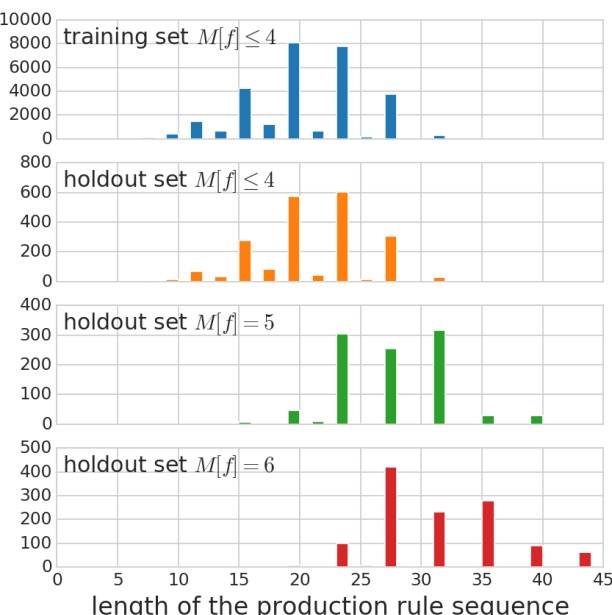

Figure A.1: Histograms of lengths of the production rule sequence.

## B  DETAILS OF THE BASELINE MODELS

We provide more details of the baseline models we proposed to be compared with our NN model. Using the same notation as in Eq. (2), the conditional distribution of the next production rule given the partial sequence and the desired condition is denoted by

$$p(r_{t+1}|r_1, \ldots, r_t, c^{(0)}, c^{(\infty)}).$$

The baseline models are essentially different ways to approximate the conditional distribution using empirical distributions.

**Limited History (LH) ($l$)**   The conditional distribution is approximated by the empirical conditional distribution given at most the last $l$ production rules of the partial sequence and the desired condition. We derive the empirical conditional distribution from the training set by first finding all the partial sequences therein that match the given partial sequence and desired condition. Then we compute the proportion of each production rule that appears as the next production rule of the found partial sequences. The proportional is therefore the empirical conditional distribution. To avoid introducing an invalid next production rule, the empirical conditional distribution is multiplied by the production rule mask of valid next production rules, and renormalized.

$$p_{\text{LH}(l)} = \hat{p}(r_{t+1}|r_{t-l+1}, \ldots, r_t, c^{(0)}, c^{(\infty)}).$$

**Full History (FH)**   The conditional distribution is approximated by the empirical conditional distribution given the full partial sequence and the desired condition. The empirical conditional distribution is derived from the training set similarly as the LH model.

$$p_{\text{FH}} = \hat{p}(r_{t+1}|r_1, \ldots, r_t, c^{(0)}, c^{(\infty)}).$$

**Limited History No Condition (LHNC) ($l$)**   The conditional distribution is approximated by the empirical conditional distribution given at most the last $l$ production rules of the partial sequence only, where the desired condition is ignored. The empirical conditional distribution is derived from the training set similarly as the LH model.

$$p_{\text{LHNC}(l)} = \hat{p}(r_{t+1}|r_{t-l+1}, \ldots, r_t).$$

**Full History No Condition (FHNC)**   The conditional distribution is approximated by the empirical conditional distribution given the full partial sequence only, where the desired condition is ignored. The empirical conditional distribution is derived from the training set similarly as the LH model.

$$p_{\text{FHNC}} = \hat{p}(r_{t+1}|r_1, \ldots, r_t).$$

## C   DETAILS OF AGGREGATING METRICS ON DIFFERENT LEVELS

The metrics are aggregated on different levels. We compute the average of success rates, syntactic novelty rates and semantic novelty rates over all the conditions in the in-sample condition region. The out-of-sample condition region is not bounded, and hence we consider the region within $|P_{x\to0}[f]| \leq 9$ and $|P_{x\to\infty}[f]| \leq 9$. Since the average can be arbitrarily small if the boundary is arbitrarily large, instead we compute the total number of success, syntactic novelty and semantic novelty expressions over all the conditions in the out-of-sample region.

The L1-distance is not well defined for an expression which is non-terminal or $\infty$. For both cases, we specify the L1-distance as 18, which is the L1-distance between conditions $(0,0)$ and $(9,9)$.

## D   DIVERSITY OF GENERATED EXPRESSIONS

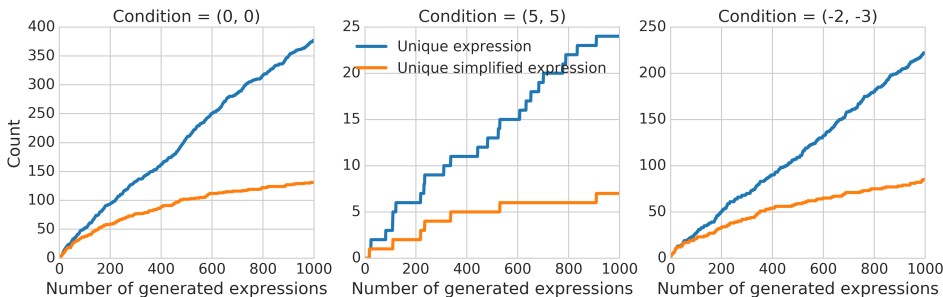

Figure D.1: **Cumulative counts of unique expressions and unique simplified expressions that satisfy the desired condition among expressions generated by the NN model on various desired conditions.** Left: desired condition $(c^{(0)}, c^{(\infty)}) = (0,0)$. Middle: desired condition $(c^{(0)}, c^{(\infty)}) = (5,5)$. Right: desired condition $(c^{(0)}, c^{(\infty)}) = (-2,-3)$.

A model that generates expressions with a high success rate (i.e., satisfying the desired condition most of the times) but lacking of diversity is problematic. To demonstrate the diversity of expressions generated by our NN model, we generate 1000 expressions on each desired condition, and compute the cumulative counts of unique expressions and unique simplified expressions that satisfy the desired condition among the first number of expressions of the 1000 generated expressions, respectively. Figure D.1 shows the cumulative counts on three typical desired conditions $(c^{(0)}, c^{(\infty)}) = (0,0), (5,5), (-2,-3)$. We can observe that the counts steadily increase as the number of generated expressions under consideration increases. Even with 1000 expressions, the counts have not been saturated.

## E   ADDITIONAL PLOTS OF METRICS FOR PRODUCTION RULE GENERATING NN AND BASELINE MODELS

Due to the space limit of the paper, Figure 4 in the main text only contains the plots of metrics on each condition for LHNC (8) and NN. Additional plots of all the baseline models in Table 3 are presented in this section. Figure E.1 contains NN, NNNC and Random. Figure E.2 contains FH and FHNC. Figure E.3 contains LH (2), LH (4), LH (8) and LH (16). Figure E.4 contains LHNC (2), LHNC (4), LHNC (8) and LHNC (16).

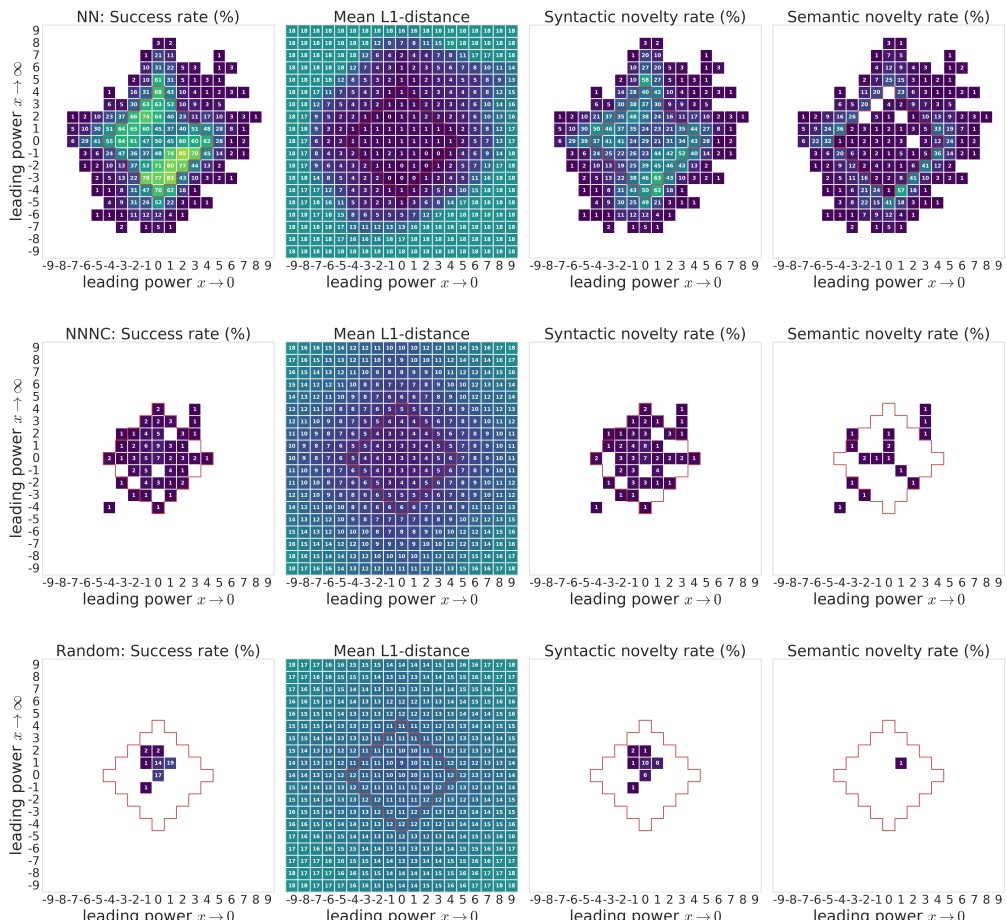

Figure E.1: **Metrics for NN, NNNC and Random models on each condition within** $|P_{x \to 0}[f]| \leq 9$ **and** $|P_{x \to \inf}[f]| \leq 9$**.** Each column corresponds to a metric: success rate, mean L1-distance, syntactic novelty rate and semantic novelty rate, respectively. Conditions with $M[f] \leq 4$ are inside the red boundary. Conditions with zero success rate, syntactic novelty rate or semantic novelty rate are left blank in the corresponding plots.

## F    SELECTION STEP IN MONTE CARLO TREE SEARCH

In the selection step, we use a variant of the PUCT algorithm (Silver et al., 2016; Rosin, 2011) for exploration,

$$U(s_t, a) = c_{\text{puct}} P(s_t, a) \frac{\sqrt{\sum_b N(s_t, b)}}{1 + N(s_t, a)}, \tag{3}$$

where $N(s_t, a)$ is the number of visits to the current node, $\sum_b N(s_t, b)$ is the number of visits to the parent of the current node, $c_{\text{puct}}$ controls the strength of exploration and $P(s_t, a)$ is the prior probability of action $a$. This strategy initially prefers an action with high prior probability and low visits for similar tree node quality $Q(s_t, a)$.

## G    EVOLUTIONARY ALGORITHM

We implemented the conventional symbolic regression approach with EA using DEAP (Fortin et al., 2012), a popular package for symbolic regression research (Quade et al., 2016; Claveria et al., 2016). We define a set of required primitives: $+, -, *, /, x$ and 1. An expression is represented as a tree where all the primitives are nodes. We start with a population of 10 individual trees with the maximum tree height set to 50. The probability of mating two individuals is 0.1, and of mutating an individual

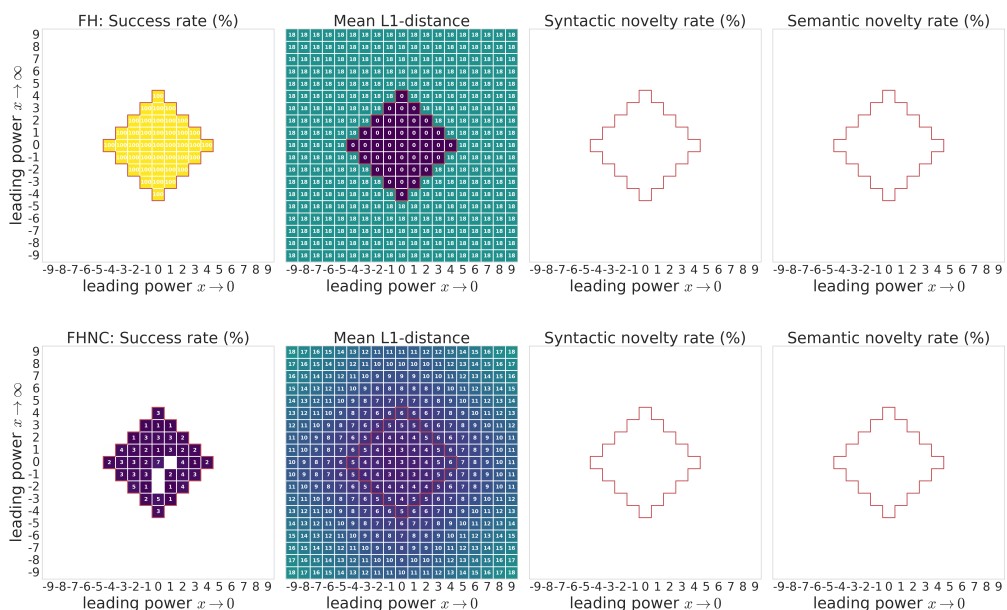

Figure E.2: **Metrics for FH and FHNC models on each condition within $|P_{x \to 0}[f]| \le 9$ and $|P_{x \to \inf}[f]| \le 9$.** Each column corresponds to a metric: success rate, mean L1-distance, syntactic novelty rate and semantic novelty rate, respectively. Conditions with $M[f] \le 4$ are inside the red boundary. Conditions with zero success rate, syntactic novelty rate or semantic novelty rate are left blank in the corresponding plots.

is 0.5 (chosen based on a hyperparameter search, see Appendix I). The limit of number of evaluations for a new offspring is set to 500 so that it is comparable to the number of simulations in MCTS.

# H    GRAMMAR VARIATIONAL AUTOENCODER

We would like to point out that the dataset in this paper is more challenging for GVAE than the dataset used in Grammar Variational Autoencoder (GVAE) paper (Kusner et al., 2017), although it is constructed with fewer production rules. First, the maximum length of production rule sequences in GVAE paper is 15, while the maximum length is 31 in our training set and 43 in holdout set. The RNN decoder usually has difficulties in learning longer sequence due to e.g. exposure bias (Bengio et al., 2015). Second, while our maximum length is more than doubled, our training set only contains 28837 expressions compared to 100000 in GVAE paper. Samples in our training set are sparser in the syntactic space than those of the GVAE paper.

We trained a GVAE using the the open-sourced code[2] with the following modifications: 1) using context-free grammar in our paper 2) setting the max length to 100 so it is comparable with MCTS and EA experiments in our paper. The GVAE paper reported reconstruction accuracy 0.537. On our validation set, the reconstruction accuracy is 0.102.

During each iteration in Bayesian optimization, 500 data in the training set are randomly selected to model the posterior distribution and the model will suggest 50 new expressions. These 50 new expressions will be evaluated and concatenated into the training set. For each symbolic regression task, we ran 5 iterations. The only difference from the setting in Kusner et al. (2017) is that instead of averaging over 10 repetitions, we average over 2 repetitions so the total number of evaluation is $2 \times 5 \times 50 = 500$ for searching each expression, which is comparable to the setting of MCTS and EA.

Due to the restriction on the computational time of Bayesian optimization for GVAE, we evaluate GVAE on a subset of holdout sets. 30 expressions are randomly selected from each of the holdout

---

[2]https://github.com/mkusner/grammarVAE

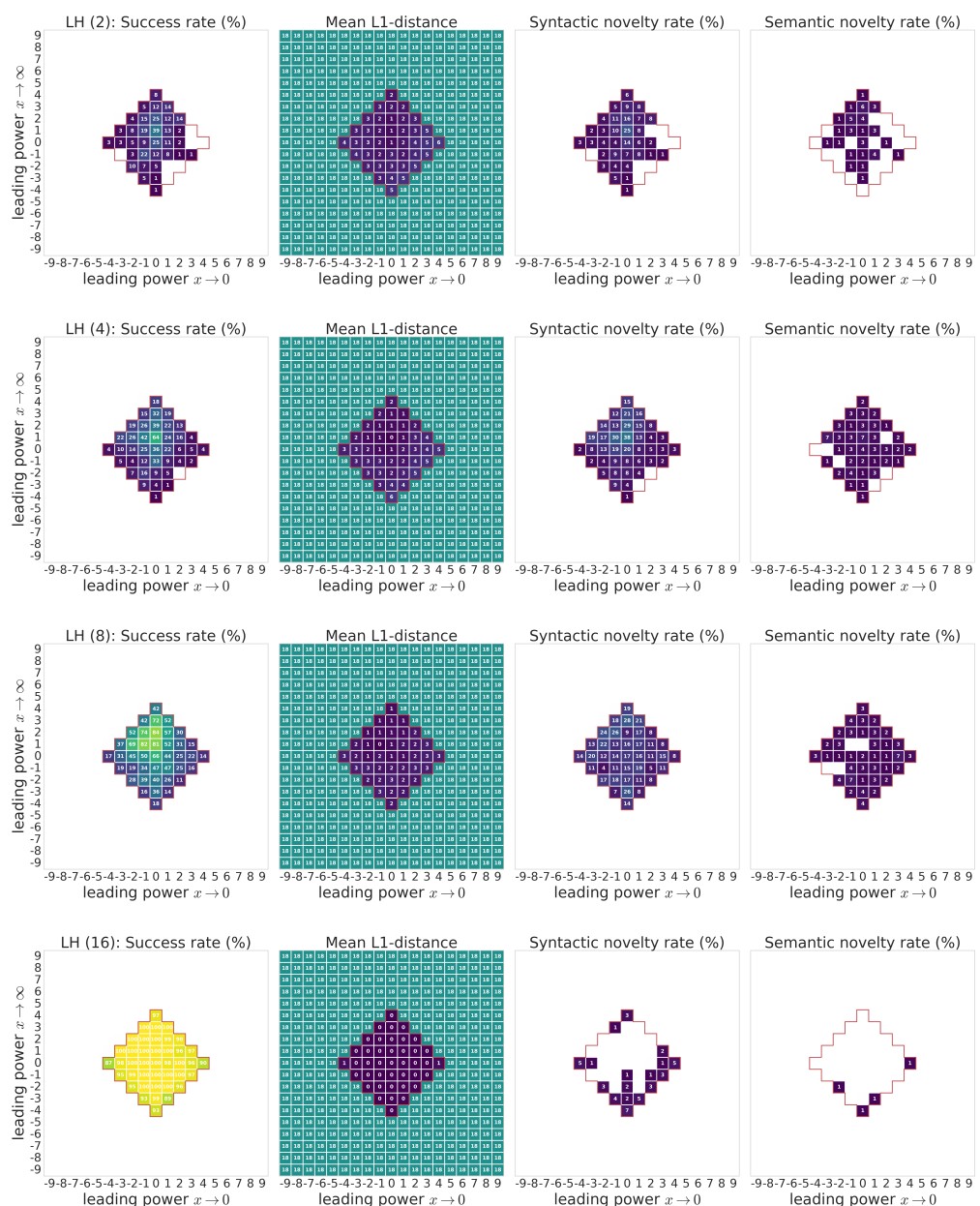

Figure E.3: **Metrics for LH models with different history length on each condition within** $|P_{x\to 0}[f]| \le 9$ **and** $|P_{x\to\inf}[f]| \le 9$. Each column corresponds to a metric: success rate, mean L1-distance, syntactic novelty rate and semantic novelty rate, respectively. Conditions with $M[f] \le 4$ are inside the red boundary. Conditions with zero success rate, syntactic novelty rate or semantic novelty rate are left blank in the corresponding plots.

sets ($M <= 4, = 5, = 6$) in Section 5 and symbolic regression tasks are performed on these 120 expressions. We report the results of NG-MCTS, GVAE and GVAE+PW (including the error of leading powers to the score function in Bayesian optimization) evaluated on median extrapolation RMSE and absolute error of leading powers in Table H.1. GVAE + PW has better $\Delta P[g]$ comparing to GVAE. NG-MCTS significantly outperforms GVAE and GVAE + PW on solved percentage and extrapolation error $\Delta g_{\text{ext.}}$.

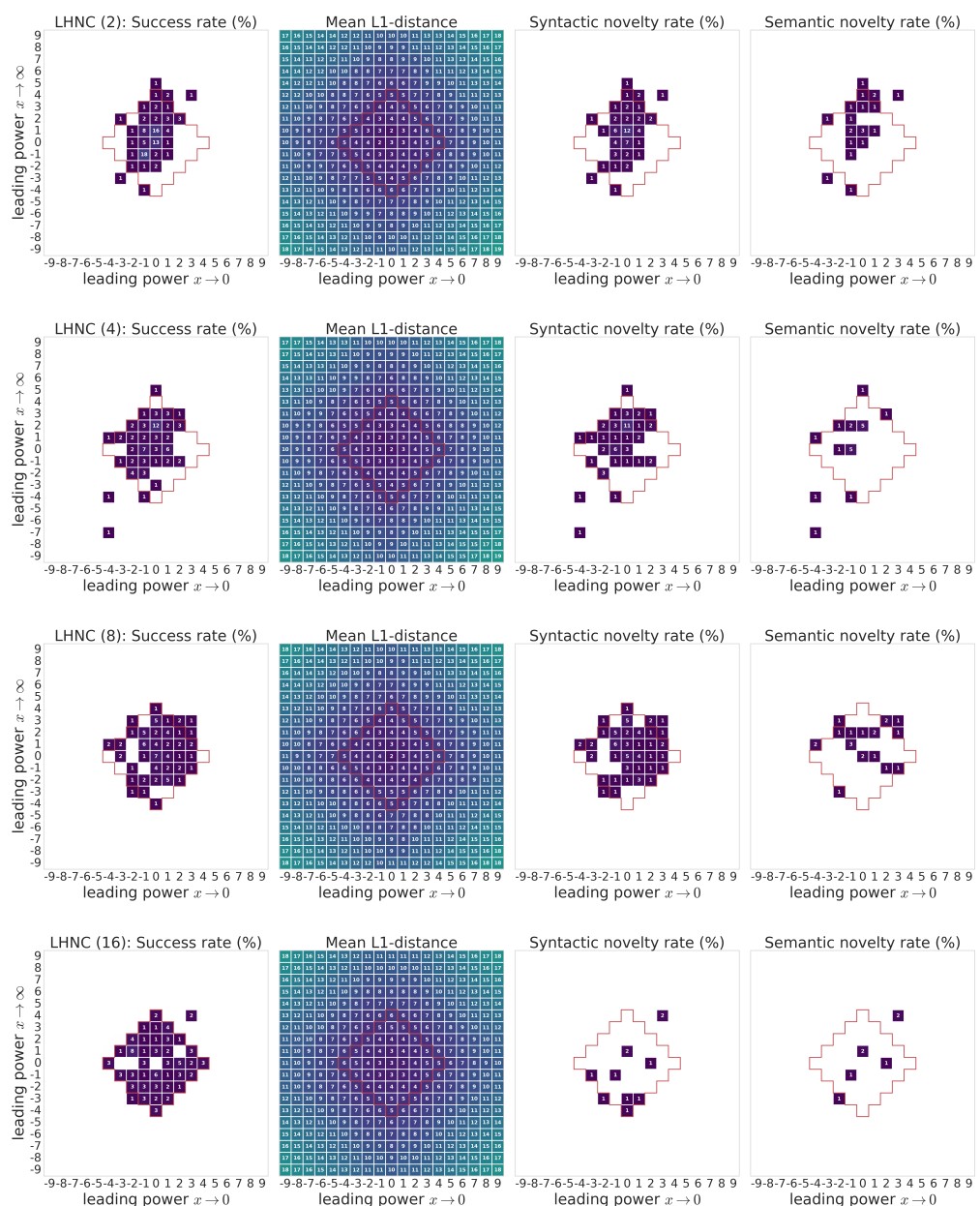

Figure E.4: **Metrics for LHNC models with different history length on each condition within** $|P_{x \to 0}[f]| \leq 9$ **and** $|P_{x \to \inf}[f]| \leq 9$. Each column corresponds to a metric: success rate, mean L1-distance, syntactic novelty rate and semantic novelty rate, respectively. Conditions with $M[f] \leq 4$ are inside the red boundary. Conditions with zero success rate, syntactic novelty rate or semantic novelty rate are left blank in the corresponding plots.

## I CHOICE OF HYPERPARAMETERS

NN is trained with batch size 256 for $10^7$ steps. The initial learning rate is $0.001$. It decays exponentially every $10^5$ steps with a base of 0.99.

The hyperparameters of MCTS (exploration strength = 50) and EA (the probability of mating two individuals = 0.1, the probability of mutating an individual = 0.5) are selected by hyperparameter searching to maximize the solved percentage in holdout set $M[f] \leq 4$ with both $\mathcal{D}_{\text{train}}$ and $P_{x \to 0}[f]$, $P_{x \to \infty}[f]$ provided.

Table H.1: Comparison with GVAE on subset of holdout sets.

| $M[f]$ | Method | Neural Guided | Objective Function | | Solved Percent | Invalid Percent | Unsolved | | | | |
|---|---|---|---|---|---|---|---|---|---|---|---|
| | | | $\mathcal{D}_{\text{train}}$ | $P_{x\to 0,\infty}[f]$ | | | Percent | $\Delta g_{\text{train}}$ | $\Delta g_{\text{int.}}$ | $\Delta g_{\text{ext.}}$ | $\Delta P[g]$ |
| $\leq 4$ | NG-MCTS | $\checkmark$ | $\checkmark$ | $\checkmark$ | **83.33%** | **0.00%** | 16.67% | 0.436 | 1.000 | **0.009** | **0** |
| | GVAE | – | $\checkmark$ | $\times$ | 10.00% | 0.00% | 90.00% | **0.217** | **0.159** | 0.599 | 2 |
| | GVAE + PW | – | $\checkmark$ | $\checkmark$ | 10.00% | 0.00% | 90.00% | 0.386 | 0.324 | 0.056 | **0** |
| $= 5$ | NG-MCTS | $\checkmark$ | $\checkmark$ | $\checkmark$ | **60.00%** | **0.00%** | 40.00% | **0.088** | **0.086** | **0.011** | **0** |
| | GVAE | – | $\checkmark$ | $\times$ | 0.00% | 0.00% | 100.00% | 0.233 | 0.259 | 0.164 | 4 |
| | GVAE + PW | – | $\checkmark$ | $\checkmark$ | 3.33% | 0.00% | 96.67% | 0.649 | 0.565 | 0.383 | 2 |
| $= 6$ | NG-MCTS | $\checkmark$ | $\checkmark$ | $\checkmark$ | **10.00%** | **0.00%** | 90.00% | 0.306 | 0.266 | **0.009** | **0** |
| | GVAE | – | $\checkmark$ | $\times$ | 0.00% | 0.00% | 100.00% | **0.260** | **0.206** | 0.037 | 5 |
| | GVAE + PW | – | $\checkmark$ | $\checkmark$ | 0.00% | 0.00% | 100.00% | 0.595 | 0.436 | 0.087 | 3 |

## J EXAMPLES OF SYMBOLIC REGRESSION RESULTS

In this section, we select expressions solved by NG-MCTS but unsolved by EA + PW in Table 1 as examples of symbolic regression results. Figure J.1, Figure J.2 and Figure J.3 show eight expressions with $M[f] \leq 4$, $M[f] = 5$ and $M[f] = 6$, respectively. The symbolic expressions, leading powers, interpolation errors and extrapolation errors of these 24 desired expressions $f(x)$, as well as their corresponding best expressions found by NG-MCTS, denoted by $g^{\text{NG−MCTS}}(x)$, and by EA + PW, denoted by $g^{\text{EA+PW}}(x)$, are listed in Table J.1.

We also select expressions solved by EA + PW but unsolved by NG-MCTS in Table 1 as examples of symbolic regression results. Figure J.4, Figure J.5 and Figure J.6 show eight expressions with $M[f] \leq 4$, $M[f] = 5$ and $M[f] = 6$, respectively. The symbolic expressions, leading powers, interpolation errors and extrapolation errors of these 24 desired expressions $f(x)$, as well as their corresponding best expressions found by NG-MCTS and EA + PW are listed in Table J.2.

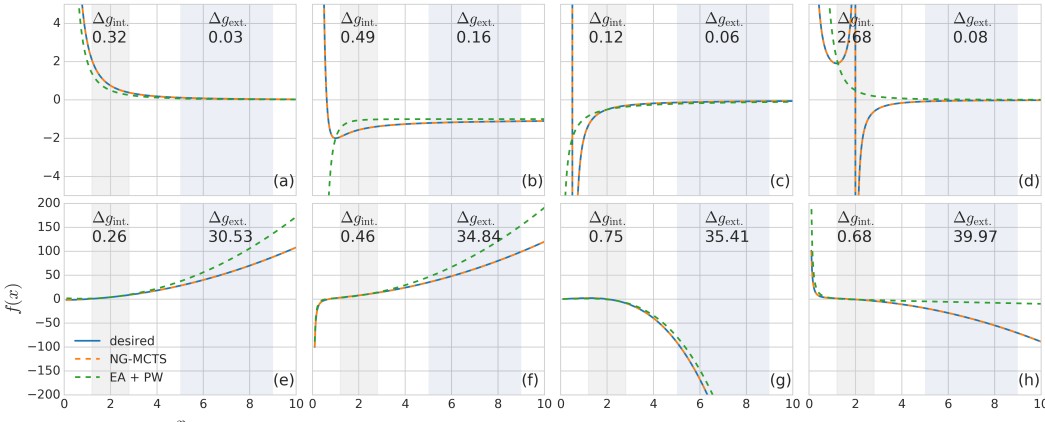

Figure J.1: **Examples of expressions solved by NG-MCTS but unsolved by EA + PW with** $M[f] \leq 4$. Each subplot of (a)-(h) demonstrates an expression solved by NG-MCTS but unsolved by EA + PW. Grey area is the region to compute interpolation error $\Delta g_{\text{int.}}$ and light blue area is the region to compute extrapolation error $\Delta g_{\text{ext.}}$. The display range of y-axis is $[-5, 5]$ for the four subplots in the first row and $[-200, 200]$ for the four subplots in the second row to show the discrepancy of expressions on two different scales.

## K SYMBOLIC REGRESSION WITH NOISE

In the main text, the training points $\mathcal{D}_{\text{train}}$ from the desired expression is noise-free. However, in realistic applications of symbolic regression, measurement noise usually exists. We add a random Gaussian noise with standard deviation $0.5$ to expression evaluations on $\mathcal{D}_{\text{train}}$ and compute $\Delta g_{\text{int.}}$ and $\Delta g_{\text{ext.}}$ using evaluations without noise. The results are summarized in Table K.1. The performance of all the methods is worse than that in the noise-free experiments, but the relative relationship

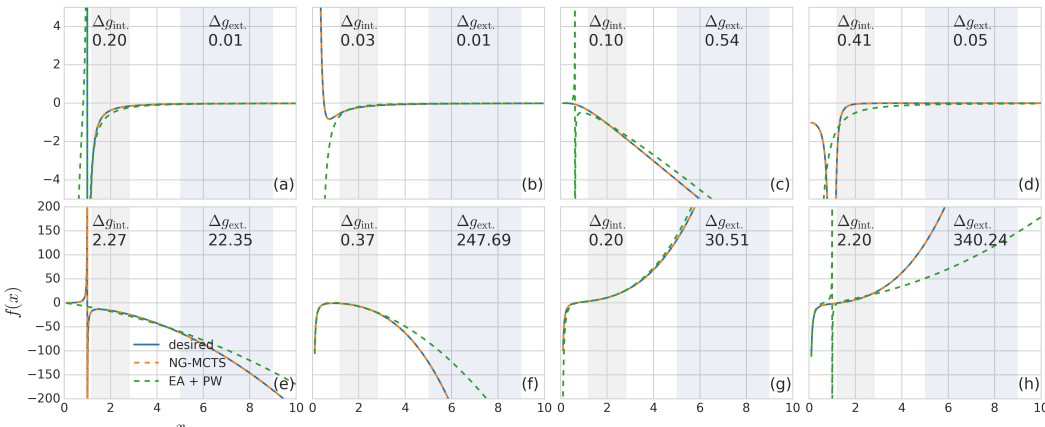

Figure J.2: **Examples of expressions solved by NG-MCTS but unsolved by EA + PW with** $M[f] = 5$. Each subplot of (a)-(h) demonstrates an expression solved by NG-MCTS but unsolved by EA + PW. Grey area is the region to compute interpolation error $\Delta g_{\text{int.}}$ and light blue area is the region to compute extrapolation error $\Delta g_{\text{ext.}}$. The display range of y-axis is $[-5, 5]$ for the four subplots in the first row and $[-200, 200]$ for the four subplots in the second row to show the discrepancy of expressions on two different scales.

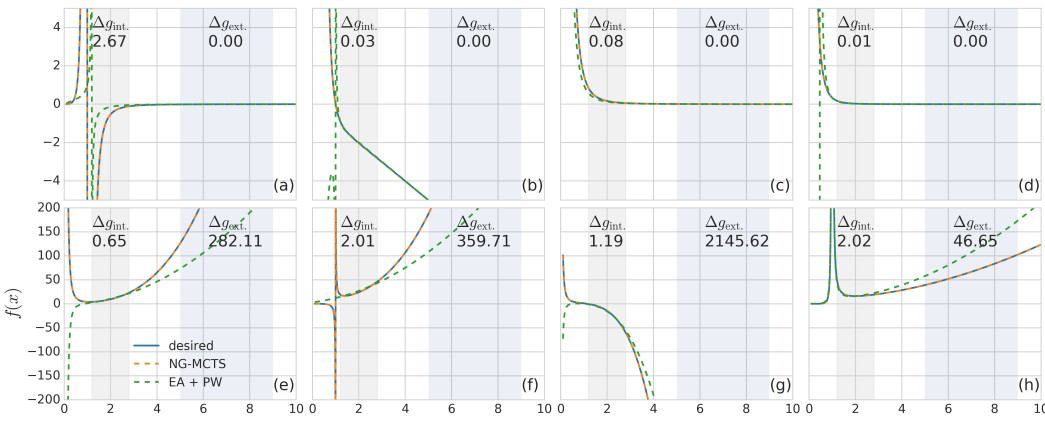

Figure J.3: **Examples of expressions solved by NG-MCTS but unsolved by EA + PW with** $M[f] = 6$. Each subplot of (a)-(h) demonstrates an expression solved by NG-MCTS but unsolved by EA + PW. Grey area is the region to compute interpolation error $\Delta g_{\text{int.}}$ and light blue area is the region to compute extrapolation error $\Delta g_{\text{ext.}}$. The display range of y-axis is $[-5, 5]$ for the four subplots in the first row and $[-200, 200]$ for the four subplots in the second row to show the discrepancy of expressions on two different scales.

still remains. NG-MCTS solves the most expressions than all the other methods. It has the lowest medians of $\Delta g_{\text{ext.}}$ and $\Delta P[g]$, suggesting good generalization in extrapolation even with noise on training points.

## L    SYNTACTIC NOVELTY EXAMPLES

For ease of presentation, we show syntactic novelties generated by NN that only have one semantically identical expression (i.e., the expression that shares the same simplified expression) in the training set. By comparing each syntactic novelty and its semantically identical expression in the training set (shown in Table L.1), we can observe that the model generates some nontrivial syntactic novelties. For

Table J.1: **Examples of expressions solved by NG-MCTS but unsolved by EA + PW.** This table shows the desired expressions $f(x)$ and their corresponding best expressions found by NG-MCTS $g^{\text{NG-MCTS}}(x)$ and by EA + PW $g^{\text{EA+PW}}(x)$. The leading powers $P_{x\to 0}[\cdot]$, $P_{x\to\infty}[\cdot]$, interpolation error $\Delta g_{\text{int.}}$ and extrapolation error $\Delta g_{\text{ext.}}$ are reported for each expression.

humans to propose such syntactic novelties, they would need to know and apply the corresponding nontrivial mathematical rules (shown in the first column of Table L.1) to derive the expressions from those already known in the training set. On the contrary, NN generates the syntactic novelties such as

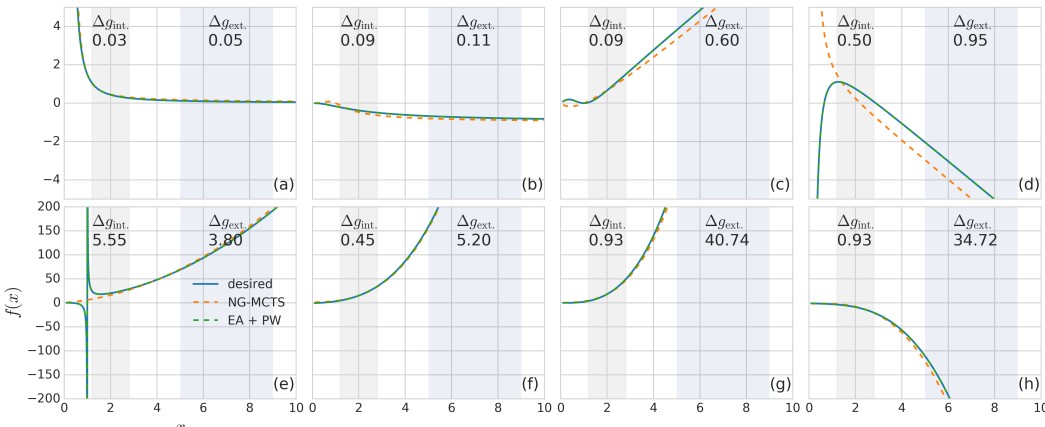

Figure J.4: **Examples of expressions solved by EA + PW but unsolved by NG-MCTS with** $M[f] \leq 4$**.** Each subplot of (a)-(h) demonstrates an expression solved by EA + PW but unsolved by NG-MCTS. Grey area is the region to compute interpolation error $\Delta g_{\text{int.}}$ and light blue area is the region to compute extrapolation error $\Delta g_{\text{ext.}}$. The display range of y-axis is $[-5, 5]$ for the four subplots in the first row and $[-200, 200]$ for the four subplots in the second row to show the discrepancy of expressions on two different scales.

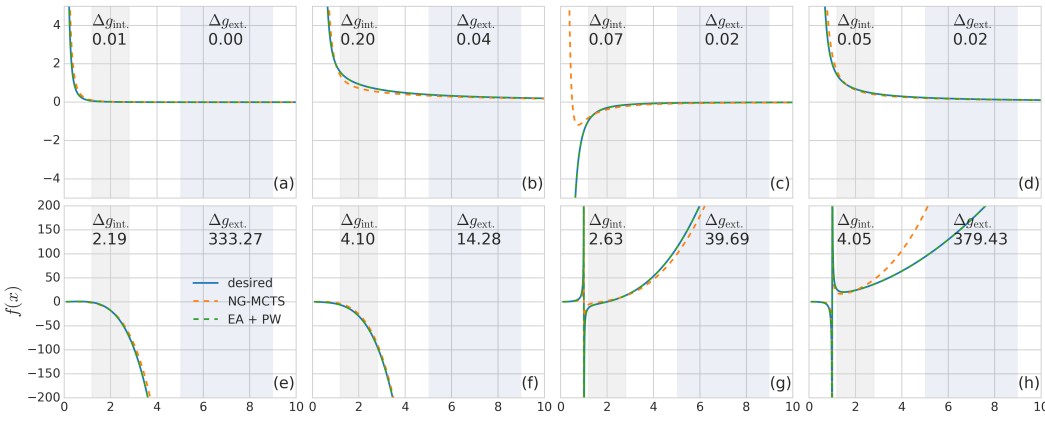

Figure J.5: **Examples of expressions solved by EA + PW but unsolved by NG-MCTS with** $M[f] = 5$**.** Each subplot of (a)-(h) demonstrates an expression solved by EA + PW but unsolved by NG-MCTS. Grey area is the region to compute interpolation error $\Delta g_{\text{int.}}$ and light blue area is the region to compute extrapolation error $\Delta g_{\text{ext.}}$. The display range of y-axis is $[-5, 5]$ for the four subplots in the first row and $[-200, 200]$ for the four subplots in the second row to show the discrepancy of expressions on two different scales.

$1/(A/B) = B/A, A/(B * C) = A/B/C$, etc, without being explicitly taught these mathematical rules.

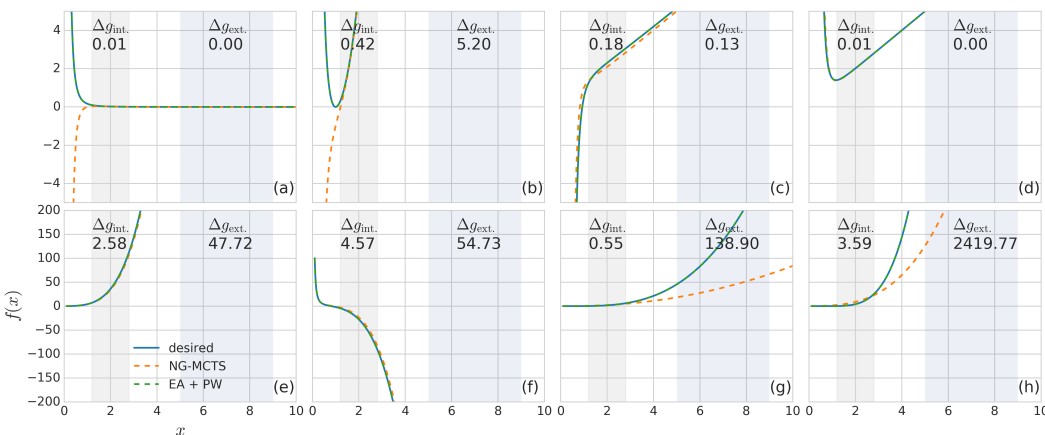

Figure J.6: **Examples of expressions solved by EA + PW but unsolved by NG-MCTS with** $M[f] = 6$. Each subplot of (a)-(h) demonstrates an expression solved by EA + PW but unsolved by NG-MCTS. Grey area is the region to compute interpolation error $\Delta g_{\text{int.}}$ and light blue area is the region to compute extrapolation error $\Delta g_{\text{ext.}}$. The display range of y-axis is $[-5, 5]$ for the four subplots in the first row and $[-200, 200]$ for the four subplots in the second row to show the discrepancy of expressions on two different scales.

Table J.2: **Examples of expressions solved by EA + PW but unsolved by NG-MCTS.** This table shows the desired expressions $f(x)$ and their corresponding best expressions found by NG-MCTS $g^{\mathrm{NG-MCTS}}(x)$ and by EA + PW $g^{\mathrm{EA+PW}}(x)$. The leading powers $P_{x\to0}[\cdot]$, $P_{x\to\infty}[\cdot]$, interpolation error $\Delta g_{\mathrm{int.}}$ and extrapolation error $\Delta g_{\mathrm{ext.}}$ are reported for each expression.

| | | Expression | $P_{x\to0}[\cdot]$ | $P_{x\to\infty}[\cdot]$ | $\Delta g_{\mathrm{int.}}$ | $\Delta g_{\mathrm{ext.}}$ |
|---|---|---|---|---|---|---|
| (a) | $f(x)$ | | -3 | -1 | — | — |
| | $g^{\mathrm{NG-MCTS}}(x)$ | | -3 | -1 | 0.026 | 0.045 |
| | $g^{\mathrm{EA+PW}}(x)$ | | -3 | -1 | 0.000 | 0.000 |
| (b) | $f(x)$ | | 3 | 0 | — | — |
| | $g^{\mathrm{NG-MCTS}}(x)$ | | 3 | 0 | 0.092 | 0.107 |
| | $g^{\mathrm{EA+PW}}(x)$ | | 3 | 0 | 0.000 | 0.000 |
| (c) | $f(x)$ | | 1 | 1 | — | — |
| | $g^{\mathrm{NG-MCTS}}(x)$ | | 1 | 1 | 0.087 | 0.602 |
| | $g^{\mathrm{EA+PW}}(x)$ | | 1 | 1 | 0.000 | 0.000 |
| (d) | $f(x)$ | | -2 | 1 | — | — |
| | $g^{\mathrm{NG-MCTS}}(x)$ | | -2 | 1 | 0.497 | 0.954 |
| | $g^{\mathrm{EA+PW}}(x)$ | | -2 | 1 | 0.000 | 0.000 |
| (e) | $f(x)$ | | 2 | 2 | — | — |
| | $g^{\mathrm{NG-MCTS}}(x)$ | | 1 | 2 | 5.554 | 3.798 |
| | $g^{\mathrm{EA+PW}}(x)$ | | 2 | 2 | 0.000 | 0.000 |
| (f) | $f(x)$ | | 0 | 3 | — | — |
| | $g^{\mathrm{NG-MCTS}}(x)$ | | 0 | 3 | 0.447 | 5.196 |
| | $g^{\mathrm{EA+PW}}(x)$ | | 0 | 3 | 0.000 | 0.000 |
| (g) | $f(x)$ | | 1 | 3 | — | — |
| | $g^{\mathrm{NG-MCTS}}(x)$ | | 1 | 3 | 0.930 | 40.741 |
| | $g^{\mathrm{EA+PW}}(x)$ | | 1 | 3 | 0.000 | 0.000 |
| (h) | $f(x)$ | | 0 | 3 | — | — |
| | $g^{\mathrm{NG-MCTS}}(x)$ | | 0 | 3 | 0.930 | 34.725 |
| | $g^{\mathrm{EA+PW}}(x)$ | | 0 | 3 | 0.000 | 0.000 |
| (a) | $f(x)$ | | -2 | -3 | — | — |
| | $g^{\mathrm{NG-MCTS}}(x)$ | | -2 | -3 | 0.013 | 0.000 |
| | $g^{\mathrm{EA+PW}}(x)$ | | -2 | -3 | 0.000 | 0.000 |
| (b) | $f(x)$ | | -4 | -1 | — | — |
| | $g^{\mathrm{NG-MCTS}}(x)$ | | -4 | -1 | 0.203 | 0.039 |
| | $g^{\mathrm{EA+PW}}(x)$ | | -4 | -1 | 0.000 | 0.000 |
| (c) | $f(x)$ | | -3 | -2 | — | — |
| | $g^{\mathrm{NG-MCTS}}(x)$ | | -3 | -2 | 0.068 | 0.020 |
| | $g^{\mathrm{EA+PW}}(x)$ | | -3 | -2 | 0.000 | 0.000 |
| (d) | $f(x)$ | | -4 | -1 | — | — |
| | $g^{\mathrm{NG-MCTS}}(x)$ | | -4 | -1 | 0.050 | 0.017 |
| | $g^{\mathrm{EA+PW}}(x)$ | | -4 | -1 | 0.000 | 0.000 |
| (e) | $f(x)$ | | 1 | 4 | — | — |
| | $g^{\mathrm{NG-MCTS}}(x)$ | | 1 | 4 | 2.190 | 333.266 |
| | $g^{\mathrm{EA+PW}}(x)$ | | 1 | 4 | 0.000 | 0.000 |
| (f) | $f(x)$ | | 1 | 4 | — | — |
| | $g^{\mathrm{NG-MCTS}}(x)$ | | 1 | 4 | 4.099 | 14.283 |
| | $g^{\mathrm{EA+PW}}(x)$ | | 1 | 4 | 0.000 | 0.000 |
| (g) | $f(x)$ | | 2 | 3 | — | — |
| | $g^{\mathrm{NG-MCTS}}(x)$ | | 2 | 3 | 2.626 | 39.693 |
| | $g^{\mathrm{EA+PW}}(x)$ | | 2 | 3 | 0.000 | 0.000 |
| (h) | $f(x)$ | | 3 | 2 | — | — |
| | $g^{\mathrm{NG-MCTS}}(x)$ | | 3 | 3 | 4.051 | 379.428 |
| | $g^{\mathrm{EA+PW}}(x)$ | | 3 | 2 | 0.000 | 0.000 |
| (a) | $f(x)$ | | -3 | -3 | — | — |
| | $g^{\mathrm{NG-MCTS}}(x)$ | | -3 | -3 | 0.012 | 0.001 |
| | $g^{\mathrm{EA+PW}}(x)$ | | -3 | -3 | 0.000 | 0.000 |
| (b) | $f(x)$ | | -3 | 3 | — | — |
| | $g^{\mathrm{NG-MCTS}}(x)$ | | -3 | 3 | 0.417 | 5.202 |
| | $g^{\mathrm{EA+PW}}(x)$ | | -3 | 3 | 0.000 | 0.000 |
| (c) | $f(x)$ | | -5 | 1 | — | — |
| | $g^{\mathrm{NG-MCTS}}(x)$ | | -5 | 1 | 0.183 | 0.127 |
| | $g^{\mathrm{EA+PW}}(x)$ | | -5 | 1 | 0.000 | 0.000 |
| (d) | $f(x)$ | | -5 | 1 | — | — |
| | $g^{\mathrm{NG-MCTS}}(x)$ | | -5 | 1 | 0.009 | 0.000 |
| | $g^{\mathrm{EA+PW}}(x)$ | | -5 | 1 | 0.000 | 0.000 |
| (e) | $f(x)$ | | 2 | 4 | — | — |
| | $g^{\mathrm{NG-MCTS}}(x)$ | | 2 | 4 | 2.582 | 47.716 |
| | $g^{\mathrm{EA+PW}}(x)$ | | 2 | 4 | 0.000 | 0.000 |
| (f) | $f(x)$ | | -2 | 4 | — | — |
| | $g^{\mathrm{NG-MCTS}}(x)$ | | -2 | 4 | 4.568 | 54.734 |
| | $g^{\mathrm{EA+PW}}(x)$ | | -2 | 4 | 0.000 | 0.000 |
| (g) | $f(x)$ | | 3 | 3 | — | — |
| | $g^{\mathrm{NG-MCTS}}(x)$ | | 3 | 2 | 0.552 | 138.902 |
| | $g^{\mathrm{EA+PW}}(x)$ | | 2 | 3 | 0.000 | 0.000 |
| (h) | $f(x)$ | | 2 | 4 | — | — |
| | $g^{\mathrm{NG-MCTS}}(x)$ | | 2 | 3 | 3.593 | 2419.773 |
| | $g^{\mathrm{EA+PW}}(x)$ | | 2 | 4 | 0.000 | 0.000 |

Table K.1: **Results of symbolic regression methods with noise.** Search expressions in holdout sets $M[f] \leq 4$, $M[f] = 5$ and $M[f] = 6$ with data points on $\mathcal{D}_{\text{train}}$ and / or leading powers $P_{x\to 0}[f]$ and $P_{x\to\infty}[f]$. The options are marked by on ($\surd$), off ($\times$) and not available (–). If the RMSEs of the best found expression $g(x)$ in interpolation and extrapolation are both smaller than $10^{-9}$ and $\Delta P[g] = 0$, it is *solved*. If $g(x)$ is non-terminal or $\infty$, it is *invalid*. *Hard* includes expressions in the holdout set which are not solved by any of the six methods. The medians of $\Delta g_{\text{train}}$, $\Delta g_{\text{int.}}$, $\Delta g_{\text{ext.}}$ and the median absolute errors of leading powers $\Delta P[g]$ for hard expressions are reported.

| $M[f]$ | Method | Neural Guided | Objective Function $\mathcal{D}_{\text{train}}$ | $P_{x\to 0,\infty}[f]$ | Solved Percent | Invalid Percent | Hard Percent | $\Delta g_{\text{train}}$ | $\Delta g_{\text{int.}}$ | $\Delta g_{\text{ext.}}$ | $\Delta P[g]$ |
|---|---|---|---|---|---|---|---|---|---|---|---|
| | MCTS | $\times$ | $\surd$ | $\times$ | 0.39% | 0.54% | | 0.689 | 0.351 | 0.371 | 3 |
| | MCTS (PW-only) | $\times$ | $\times$ | $\surd$ | 0.34% | **0.00%** | | – | 1.003 | 0.865 | 1 |
| $\leq 4$ | MCTS + PW | $\times$ | $\surd$ | $\surd$ | 0.34% | 0.49% | 71.41% | 0.887 | 0.643 | 0.825 | 2 |
| | NG-MCTS | $\surd$ | $\surd$ | $\surd$ | **24.29%** | 0.05% | | 0.432 | 0.449 | **0.256** | **0** |
| | EA | – | $\surd$ | $\times$ | 4.44% | 3.95% | | 0.399 | **0.223** | 0.591 | 3 |
| | EA + PW | – | $\surd$ | $\surd$ | 4.34% | 0.39% | | **0.385** | 0.489 | 0.260 | **0** |
| | MCTS | $\times$ | $\surd$ | $\times$ | 0.00% | 4.00% | | 0.931 | 0.599 | 0.972 | 5 |
| | MCTS (PW-only) | $\times$ | $\times$ | $\surd$ | 0.00% | **0.00%** | | – | 1.394 | 1.000 | 3 |
| $= 5$ | MCTS + PW | $\times$ | $\surd$ | $\surd$ | 0.00% | 3.00% | 83.40% | 0.944 | 0.817 | 0.727 | 5 |
| | NG-MCTS | $\surd$ | $\surd$ | $\surd$ | **10.80%** | 0.10% | | 0.558 | 0.430 | **0.103** | **0** |
| | EA | – | $\surd$ | $\times$ | 1.00% | 3.10% | | 0.480 | **0.256** | 0.266 | 4 |
| | EA + PW | – | $\surd$ | $\surd$ | 1.80% | 1.80% | | **0.448** | 0.382 | 0.122 | **0** |
| | MCTS | $\times$ | $\surd$ | $\times$ | 0.00% | 9.75% | | 0.960 | 0.762 | 0.888 | 6 |
| | MCTS (PW-only) | $\times$ | $\times$ | $\surd$ | 0.00% | **0.00%** | | – | 1.024 | 0.861 | 4 |
| $= 6$ | MCTS + PW | $\times$ | $\surd$ | $\surd$ | 0.00% | 8.83% | 78.75% | 1.122 | 0.807 | 0.163 | 4 |
| | NG-MCTS | $\surd$ | $\surd$ | $\surd$ | **10.33%** | 0.08% | | 0.426 | **0.205** | **0.009** | **0** |
| | EA | – | $\surd$ | $\times$ | 0.67% | 4.58% | | 0.463 | 0.427 | 0.852 | 5 |
| | EA + PW | – | $\surd$ | $\surd$ | 1.42% | 6.50% | | **0.388** | 0.369 | 0.065 | **0** |

Table L.1: **Examples of syntactic novelties to demonstrate what the NN model learned.** *Semidentical expression* refers to the expression in the training set that shares the same simplified expression as the corresponding syntactic novelty. The first column shows the mathematical rules a human needs to know and apply to derive each syntactic novelty from its semantically identical expression in the training set.

| MATHEMATICAL RULE | SYNTACTIC NOVELTY / SEM-IDENTICAL EXPRESSION |
|---|---|
| $1/(A/B) = B/A$ | $1 + 1/(1 + (x/(1+x)))$ / $1 + (1+x)/((1+x) + x)$ |
| $A/(B+B) = A/B/2$ | $1 - x/((1+x) + (1+x))$ / $1 - (x/(1+x))/(1+1)$ |
| $A + B = B + A$ | $x/((x/(1+x)) + x + x)$ / $x/(x + (x/(1+x)) + x)$ |
| $A/(B*C) = A/B/C$ | $(1 - (x/(1-x)))/(x + x*x)$ / $(1 - (x/(1-x)))/(x+1)/x$ |
| $A/C + B/C = (A+B)/C$ | $(1 + (1/x))/(x - (x*x) - 1)$ / $((1+x)/x)/(x - 1 - (x*x))$ |

