# OpenReview forum: "Neural-Guided Symbolic Regression with Asymptotic Constraints"
_ICLR.cc/2020/Conference — Reject_

### Official Review · AnonReviewer2 · 2019-10-22
**Official Blind Review #2**

**Rating:** 3

**Review:**

This paper considers a task of symbolic regression (SR) when additional information called 'asymptotic constraints' on the target expression is given. For example, when the groundtruth expression is 3 x^2 + 5 x, it behaves like x^2 in the limit x -> infinity, thus the power '2' is given as additional information 'asymptotic constraint'. In the paper's setting, for SR with univariate groundtruth functions f(x), 'asymptotic constraints' for x-> infinity and x -> 0 are given. For this situation, the paper proposes a method called NG-MCTS with an RNN-based generator and MCTS guided by it to consider asymptotic constraints. In SR, a learner is asked to acquire an explicit symbolic expression \hat{f}(x) from a given set of datapoints {x_i, f(x_i)}, but unlike the parameter optimization aspect of a standard supervised ML setting, the problem is essentially combinatorial optimization over exponentially large space of symbolic expressions of a given context-free grammar (CFG). For a given symbolic space with a prespecified CFG, extensive experimental evaluations are performed and demonstrated significant performance gain over existing alternatives based on EA. Also, quantitative evaluations about extrapolative performance and detailed evaluation of the RNN generator are also reported.

Though it includes a lot of quite interesting methods and results, the paper has two major issues:

(1) the proposed method NG-MCTS explicitly uses the fact that the target expressions are generated from a CFG, but this assumption sounds unnatural if the target problem is SR (unlike the case of GVAE). Apparently, all results except a toy case study in 5.2 depend on artificial datasets from a CFG, and any practical advantages or impacts are still unclear because the experimental settings are very artificial.

(2) the most important claim of this paper would be the proposal to use 'asymptotic constraints', but the availability of this information sounds too strong and again artificial in practical situations. A one-line motivation saying that 'this property is known a priori for some physical systems before the detailed law is derived' is not convincing enough.

**Experience Assessment:**

I do not know much about this area.

**Review Assessment: Checking Correctness Of Derivations And Theory:**

N/A

**Review Assessment: Checking Correctness Of Experiments:**

I assessed the sensibility of the experiments.

**Review Assessment: Thoroughness In Paper Reading:**

I read the paper at least twice and used my best judgement in assessing the paper.

---

> ### Author Response · Authors · 2019-11-08
> **Response to Official Blind Review #2**
>
> Thank you for your review and comments.
>
> (1) Any symbolic regression method requires to define a search space of mathematical expressions. The search space includes mathematical building blocks such as algebraic operators {+, -, *, /}, variables and constants. Using CFG is one common way to define such an infinite search space. We would like to note that the evolutionary algorithm based symbolic regression also can only search within the search space. If the search search does not have {*, /}, it will not be able to find the target expression with {*, /}.
>
> Depending on the algorithm, the expression can be represented in different ways. For example, Schmidt & Lipson (2009) represents an expression as a graph where the operators are nodes. GVAE (Kusner et al., 2017) represents an expression as a tree of CFG production rules. Any sets of operators in evolutionary algorithm based symbolic regression can be represented by CFG.
>
> We improved the presentation to make it more explicit why we train a neural network before the MCTS based search, while most symbolic regression algorithms search for expressions directly.
>
>     (a) We only sample a small fraction of expressions in the space to train the network. The network is designed to learn the relation between syntactic structure and leading powers, which can generalize to larger space and higher leading powers. The network is not designed to memorize the expressions in the space. Actually, memorization hurts the generalization tremendously outside of training space, see Full History (FH) and Full History No Condition (FHNC) in Table 3.
>     (b) Our experiments show that our method can generalize to unseen leading powers (M[f] > 4) in larger space than the space considered during training (M[f] <= 4). The neural network can guide the search efficiently even when the target expression is not in the space of training expressions.
>
> Unlike previous works in symbolic regression that only evaluate the technique on few tens of mathematical expressions (e.g. 1 in the case of GVAE), we evaluate our technique on thousands of expressions of varying complexity both with conditions seen in training (M[f]<=4) and conditions that do not exist in training set (M[f]>4). We hope our large evaluation benchmark dataset would become a more standard practice in symbolic regression community, so that one can better evaluate the benefits and shortcomings of different techniques.
>
> (2) Although using asymptotic constraints to derive mathematical formulas is common for physicists and mathematicians, it may not be as familiar for researchers in machine learning community. Besides the heat capacity (T^3) and gravitational field (1/r) examples in the introduction, there are many more real-world practical settings where such information is available. For example,
>     * London dispersion force (https://en.wikipedia.org/wiki/London_dispersion_force) is a force between atoms and molecules (with force decreasing with separation R as 1/R^3 or 1/R^6 depending on the type of matters). The expression of it is constructed by the asymptotic decay of the interactions.
>     * In density functional theory, Kato theorem (https://en.wikipedia.org/wiki/Kato_theorem) states that the electron density has a cusp at the position of the nuclei. This asymptotic behavior at zero radius is essential to construct the density and density functional.
>     * In differential equations, asymptotic analysis is used to get the trial solutions and to determine some of the coefficients.
>     * In defining kernels for stationary Gaussian random fields, the value of kernel need to decay to zero when the distance of two locations increases to infinity.
>
> The key idea of our paper is to present the first mechanism to condition the expression generating model with additional information, of which “leading power” is an example in the symbolic regression domain, and then using it to guide search more effectively in MCTS. As discussed in the paper, we believe this framework can generalize to additional expression properties, as well as other domains such as program synthesis where such properties might include the expected time complexity of the desired programs or maximum loop nesting.
>
> We hope the above clarifications addressed your concerns and we will be happy to discuss more if there are more questions or comments.

---

> > ### Author Response · Authors · 2019-11-13
> > **Response to Official Blind Review #2**
> >
> > Dear reviewer,
> >
> > Please let us know if the response helped address the two major issues and if there might be any additional questions or clarifications needed for reconsideration.

---

> > > ### Comment · AnonReviewer2 · 2019-11-14
> > > **On points (1) and (2)**
> > >
> > > Thank you for the response.
> > > I would like to make sure of my two points in other words.
> > >
> > > I guess that the technical combinations NN + MCTS for the generative tasks assuming CFGs are not new, and examples are GVAE (Kusner et al., 2017) and following works like syntax-directed variational autoencoder (SD-VAE) (Dai+ ICLR 2018). These assume CFGs because the target problem is generating (grammatically valid) SMILES strings representing chemical molecules. In these lines of research, we have many other examples (For example, see a review paper in Science (2018) DOI: 10.1126/science.aat2663).
> > >
> > > So the points of the paper would be (1) applications of these strategies to symbolic regression (SR) tasks, and (2) taking into account additional 'asymptotic constraints' in SR tasks. My points (1) and (2) also correspond to these two.
> > >
> > > For (1), I raised a question about how valid applying these grammar-based methods to symbolic regression tasks (that try to uncover hidden natural laws presumably?) is, whereas situations that GVAE and SD-VAE assumed had natural motivations and real-world datasets. Experiments on synthetic datasets would provide limited practical values on any representation learning on these hidden natural laws. (Any real-world datasets available?)
> > >
> > > For (2), I understood these include many examples, but again, can this constraint be validated using any not-synthetic datasets? Adding more information on the ground truth improves prediction performance would be not surprising, and how can we get such information when we try to apply SR to "yet unseen natural laws" where it would be hard to assume anything beforehand? In this paper's experiments, we can consider these 'asymptotic constraints' because the datasets are synthetic and we already know the ground truths. This sounds a bit like a HARKing (hypothesizing after the results are known) situation.
> > >
> > > I (groundlessly...) assumed that SR is not so familiar for the ICLR readers, and making clear these validities would help us to understand the paper's scientific contributions.

---

> > > > ### Author Response · Authors · 2019-11-15
> > > > **Novelty and relation to previous work**
> > > >
> > > > We must respectfully disagree with “NN + MCTS for the generative tasks assuming CFGs are not new”.  Both GVAE and SD-VAE (and other work) use autoencoder + Bayesian optimization, while we are using NN + MCTS. The optimization and generation procedure is quite different, with different loss functions and training procedures.
> > > >
> > > >     * The autoencoder is used to generate the entire expression but our NN generates probability distribution over production rules. We have shown that it improves generalization.
> > > >     * MCTS can search larger space of expressions compared to the search space of training expressions.
> > > >
> > > > We agree that the use of asymptotic constraints is novel and no other method is currently able to use them. We provided several examples from physical systems where asymptotic behavior can be derived even when the full form is not known.

---

> > > > ### Author Response · Authors · 2019-11-15
> > > > **Why CFG and grammar-based methods are valid for symbolic regression tasks?**
> > > >
> > > > The task of discovering “hidden natural laws” is somewhat ill-defined. The precise task of symbolic regression that has been defined and worked on for decades is to search for a small mathematical expression that fits observed data (primarily input/output pairs of an unknown function).
> > > >
> > > > CFGs are one of the most natural ways to represent mathematical expressions as is also done in many previous symbolic regression techniques. As you may recall, the first task in the GVAE paper is in fact a symbolic regression task (on a single artificial expression 1/3 + x + sin(x ∗ x) ). That’s why we and other previous works use a CFG for symbolic regression.

---

> > > > ### Author Response · Authors · 2019-11-15
> > > > **Why use synthetic datasets?**
> > > >
> > > > As with previous symbolic regression techniques (see for example [1-3]), we evaluate NG-MCTS on synthetic benchmarks. Creating and using synthetic datasets is a well established procedure for methods development. Such synthetic datasets allow the complexity of the problem to be systematically varied and techniques to be compared in rigorous and more statistically sound ways. If you consider examples where symbolic regression has been applied to real problems (e.g. [4, 5]), only a couple of expressions can be considered. This tiny size makes it difficult to confidently compare different methods.

---

> > > > > ### Author Response · Authors · 2019-11-15
> > > > > **references**
> > > > >
> > > > > [1] Kusner, Matt J., Brooks Paige, and José Miguel Hernández-Lobato. "Grammar variational autoencoder." Proceedings of the 34th International Conference on Machine Learning-Volume 70. JMLR. org, 2017.
> > > > > [2] Uy, Nguyen Quang, et al. "Semantically-based crossover in genetic programming: application to real-valued symbolic regression." Genetic Programming and Evolvable Machines 12.2 (2011): 91-119.
> > > > > [3] Arabshahi, Forough, Sameer Singh, and Animashree Anandkumar. "Combining Symbolic Expressions and Black-box Function Evaluations in Neural Programs." arXiv preprint arXiv:1801.04342 (2018).
> > > > > [4] Schmidt, Michael, and Hod Lipson. "Distilling free-form natural laws from experimental data." science 324.5923 (2009): 81-85.
> > > > > [5] Ouyang, Runhai, et al. "Simultaneous learning of several materials properties from incomplete databases with multi-task SISSO." Journal of Physics: Materials 2.2 (2019): 024002.

---

> > > > ### Author Response · Authors · 2019-11-15
> > > > **Is asymptotic constraints a HARKing?**
> > > >
> > > > First, although it may not be surprising that adding more information improves the performance. But trivially adding it in the objective function does not help. Adding this additional information to previous state-of-the-art symbolic regression techniques like evolutionary algorithm and GVAE leads to poor results. Even adding it directly to MCTS performs worse. One key contribution of our paper is to develop a new mechanism to do it efficiently. We show that our technique works significant well compared to previous state of the art symbolic regression techniques (GVAE, evolutionary techniques, MCTS).
> > > >
> > > > Second, in the paper, for a family of rational expressions, we use “leading power” as an example of the global property that is difficult to incorporate by hard constraint on local syntactic structure or numerical points. This kind of property is very helpful for expression discovery in natural sciences by human experts (see list of examples in the previous response). But there is no efficient way to incorporate this kind of knowledge in symbolic regression. Rather than hypothesizing after the results are known, we present a mechanism to successfully incorporate such knowledge.
> > > >
> > > > We agree with the reviewer that we would like to enable scientists to discover new laws using our framework, but we believe NG-MCTS is already a significant improvement over state of the art symbolic regression techniques on thousands of complex regression tasks.

---

### Official Review · AnonReviewer3 · 2019-10-24
**Official Blind Review #3**

**Rating:** 3

**Review:**

Summary:
The authors of this paper propose a novel approach for symbolic regression. The simulation results demonstrate that the proposed approach can find a better function g producing similar results as desired function f by providing the additional information – the leading power of function f.

Paper strength:
1.	The proposed NG-MCTS is elegant to solve the problem of symbolic regression.
2.	Experiment results illustrate the superiority of the proposed approach

Paper weakness:

1.	I can follow most of the mathematics in the paper. But the most confusing part for me is why you feed the random partial sequence for training. Besides, how you do inference to generate a sequence of the production rules.
2.	What is the final objective function? If I do not misunderstand, it could be the cross-entropy loss between the output of GRU and the next target production rule, RMSE and the error on leading power. Then how you optimize it? Please describe more details about this.
3.	The authors of this paper introduce more information – leading power of desired function for symbolic regression but they incorporate the additional information by introducing a simple loss function term. How about the performance of baseline approaches with those kinds of information?
4.	The whole systems seem like very complicate and it would be more interesting if the authors provide sufficient ablations to decompose the complex algorithm.


**Experience Assessment:**

I do not know much about this area.

**Review Assessment: Checking Correctness Of Derivations And Theory:**

N/A

**Review Assessment: Checking Correctness Of Experiments:**

I assessed the sensibility of the experiments.

**Review Assessment: Thoroughness In Paper Reading:**

I read the paper at least twice and used my best judgement in assessing the paper.

---

> ### Author Response · Authors · 2019-11-08
> **Response to Official Blind Review #3**
>
> Thanks for your feedback and questions. We apologize for the confusion on the framing of this work and appreciate the opportunity to clarify.
>
> There are two main components in our framework:
>
> The first component is a neural network that is trained to generate production rules to construct expressions that are consistent with a given pair of leading powers. This network learns the relation between syntactic structure and semantic prior (leading powers). Note that the numerical points are not used in this component.
>
> The second component is the MCTS algorithm that searches for an expression consistent with a given set of input-output numerical points. This second component uses the neural network from the first component to guide the search process more effectively.
>
> 1. The neural network does not generate expression directly. It is used to guide the search algorithm to the subspace with desired property (leading powers in this case).  The training dataset of (partial sequence, leading power, next production rule) is randomly sampled from the full production rule sequences of 28837 expressions, and contains the relation between syntactic structure and leading powers. Since we use a small fraction of expressions in the space for training the network, random sampling ensures that we obtain samples that best represent the distributions of possible combinations of (partial sequence, leading power, next production rule).
>
> For inference, the neural network generates production rules auto-regressively to construct expressions, by predicting a sequence of production rules starting from the start symbol until all the leaf nodes in the constructed parse tree of the expression are terminal (as described in the first paragraph in Section 5.3). The network favors the subspace of expressions with desired leading powers. This results in making MCTS more efficient to find expressions that fit the data points and have the desired leading powers.
>
> 2. For training the neural network, the training objective is minimizing the cross-entropy loss. We have a training set of (leading powers, expression) dataset, which is used to construct the corresponding dataset of (partial sequence, leading powers, next production rule). The network takes as input the leading powers and the sequence of production rules corresponding to the partial expression, and learns to generate the next production rule. It is formulated as a multi-class classification problem.
>
> After training the neural network, we use MCTS for searching the expression that fits a given set of data points. It tries to minimize the RMSE on training points and the absolute difference between the desired leading powers and leading powers of the current expression as the objective function.
>
> 3. We agree with the importance of providing a fair set of baselines.
>
> Table 1 includes baseline approaches with the additional leading power information (MCTS+PW, EA+PW, and GVAE+PW). NG-MCTS (71.22% solved) significantly outperforms these baselines with the same information: EA+PW (23.27%), MCTS+PW (0.2%) and GVAE+PW (10%) for M[f]<=4, and similarly for other cases with M[f]=5 and M[f]=6.
>
> 4. Ablations can provide important insight into how to attribute the improved performance. We included several ablations in the experiments on the two main components in our framework:
>
>     (a) For the neural network that learns to generate grammar production rules conditioned on leading powers, we evaluate models with No Condition (NNNC), Random, Full History (FH), Full History No Condition (FHNC), Limited History (LH), and Limited History No Condition (LHNC) in Section 5.3.
>
>     (b) For the MCTS, we compare MCTS with full information (points and leading power) , MCTS with only leading power information (MCTS (PW-only)),  and MCTS with only the data points. We perform similar ablations for EA and GVAE.
>
> All these ablations show the importance of training the neural network on leading powers, and the MCTS that is guided by the neural network model.
>
> We hope the above clarifications addressed your concerns and we will be happy to discuss more if there are more questions or comments.

---

> > ### Author Response · Authors · 2019-11-13
> > **Response to Official Blind Review #3**
> >
> > Dear reviewer,
> >
> > Please let us know if the response helped address the questions about the overall training and inference setup, and the strong baseline and ablation experiments. If not, could you please let us know if there might be any additional questions or clarifications needed for reconsideration.

---

### Official Review · AnonReviewer1 · 2019-10-25
**Official Blind Review #1**

**Rating:** 8

**Review:**

Summary:

This paper introduces the use of asymptotic constraints to find
prune the search space of mathematical expressions for symbolic
regression.  This is done by training a neural network to
generate production rules conditioned on being given the
asymptotic constraints and previously generated production
rules. This neural network is then itself also used to guide a
MCTS to generate mathematical expressions.

The algorithms is compared against a reasonable set of baselines
and related algorithms.

Feedback:

This is a very clear and well-written paper. It was
straightforward to understand and very easy to see how it fits in
with the broader literature. The insight about using asymptotic
constraints makes the result a bit limited to only generating
mathematical expressions, and it would have been a bit nicer if
there was something more generically applicable to program
synthesis in general. It's not really clear to me how the
existing work extends to programs.

The evaluation was very thorough and the appropriate algorithms
were compared against the work. I came away with a good
understanding of how well the model generalizes to larger
expressions compared to existing work.

Minor notes:

The abstract is a bit inaccurate as the NN generates production rules
and not expressions.

**Experience Assessment:**

I have read many papers in this area.

**Review Assessment: Checking Correctness Of Derivations And Theory:**

N/A

**Review Assessment: Checking Correctness Of Experiments:**

I carefully checked the experiments.

**Review Assessment: Thoroughness In Paper Reading:**

N/A

---

> ### Author Response · Authors · 2019-11-08
> **Response to Official Blind Review #1**
>
> Thank you for your encouraging comments and constructive feedback.
>
> We presented a general two-step framework of first learning a generative model of production rules in a grammar and then using that model to guide MCTS for efficient search. We acknowledge that we evaluated this framework in the context of symbolic regression and focused on the leading power properties. However, we believe our framework and similar modeling ideas could be equally useful in general program synthesis settings, where other properties such as program’s desired time complexity or maximum control flow nesting could be used to condition the generative model. We hope to explore such directions in future, but we will add more discussion about the generality of our approach.
>
> We agree that the naming of the neural network component is confusing. We have changed the name to "conditional production rule generating neural network" everywhere and clarified it in the paper, including in the abstract.

---

### Author Response · Authors · 2019-11-08
**Feedback Incorporated In New Paper Version**

We thank all the reviewers for their valuable and constructive feedback.

We have uploaded a new paper version to address the comments and feedback:

* According to the suggestion from Reviewer #1, we changed the name of the neural network component to "conditional production rule generating neural network" and clarified it in the paper.

* Added discussion on how similar idea of this two-step framework could be useful in other fields, e.g. program synthesis, in the future work (Conclusion).

---

### Decision · Program_Chairs · 2019-12-19

**Decision:**

Reject

**Comment:**

This paper proposes 1) using neural-guided Monte-Carlo Tree Search to search for expressions that match a dataset and 2) Augments the loss to match the asymptotics of the true function when these are given.

The use of MCTS sounds more sensible than standard evolutionary search.  The augmented loss could make sense but seems extremely niche, requiring specific side information about the problem being solved.

Overall, the task is so niche that I don't think it'll be of wide interest.  It's not clear that it's solving a real problem.